# Identification of a new stem cell population that generates *Drosophila* flight muscles

**Rajesh D Gunage[1], Heinrich Reichert[2], K VijayRaghavan[1]\***

[1]National Centre for Biological Sciences, Tata Institute of Fundamental Research, Bangalore, India; [2]Biozentrum, University of Basel, Basel, Switzerland

**Abstract** How myoblast populations are regulated for the formation of muscles of different sizes is an essentially unanswered question. The large flight muscles of *Drosophila* develop from adult muscle progenitor (AMP) cells set-aside embryonically. The thoracic segments are all allotted the same small AMP number, while those associated with the wing-disc proliferate extensively to give rise to over 2500 myoblasts. An initial amplification occurs through symmetric divisions and is followed by a switch to asymmetric divisions in which the AMPs self-renew and generate post-mitotic myoblasts. Notch signaling controls the initial amplification of AMPs, while the switch to asymmetric division additionally requires Wingless, which regulates Numb expression in the AMP lineage. In both cases, the epidermal tissue of the wing imaginal disc acts as a niche expressing the ligands Serrate and Wingless. The disc-associated AMPs are a novel muscle stem cell population that orchestrates the early phases of adult flight muscle development.

## Introduction

Stem cells populations can expand exponentially through symmetric division or asymmetrically divide to self-renew and produce a daughter cell, which differentiates to contribute to tissue formation or regeneration (*Morrison and Kimble, 2006*; *Micchelli and Perrimon, 2006*; *Mandal et al., 2007*; *Maurange et al., 2008*; *Knoblich, 2001*; *Egger et al., 2008*; *Doe, 2008*; *Sousa-Nunes et al., 2010*; *Brand and Livesey, 2011*). Many mature tissues such as colon, prostate, lung, muscle and brain use adult-specific stem cells in tissue maintenance or regeneration (*Reya et al., 2001*; *Gonzalez, 2007*). The molecular mechanisms of regulating stem-cell proliferation have been studied in neural-, intestinal-, hematopoietic- and epithelial-stem cells of vertebrates and invertebrates (*Tulina and Matunis, 2001*; *Barker et al., 2007*; *Ohlstein and Spradling 2007*; *Takashima et al., 2008*; *Farkas and Hutter, 2008*; *Reichert, 2011*; *Homem and Knoblich 2012*). The regulation of stem-cell proliferation is usually through signals from a 'stem-cell niche' (*Mandal et al., 2007*; *Brack et al., 2008*; *Chen et al., 2012*; *Cordero et al., 2012*). Significant progress in understanding these mechanisms of regulation of stem-cell proliferation and self-renewal has been made in *Drosophila*. For example, in the developing optic lobe, stem cell-like neuroepithelial cells first increase in number through symmetric divisions and are then transformed into so-called neuroblasts which undergo self-renewing asymmetric divisions to generate differentiated neurons and glial cells in this part of the brain (*Maurange et al., 2008*; *Egger et al., 2011*). Recent work has identified signaling pathways and niches required for stem cell proliferation (*Egger et al., 2010*; *Ngo et al., 2010*; *Reddy et al., 2010*; *Orihara-Ono et al., 2011*; *Wang and Rudnicki, 2011*).

*Drosophila* flight muscles are formed from adult muscle precursors (AMPs) (*Currie and Bate, 1991*; *Fernandes et al., 1991*; *Roy and VijayRaghavan, 1999*). Myogenesis occurs in two phases; an embryonic one, which makes the muscles required for the larval life (*Bate et al., 1991*) while a postembryonic phase leads to formation of muscle required for the adult (*Fernandes et al., 1991*; *Roy and VijayRaghavan, 1998*; *Sudarsan et al., 2001*). The AMPs, lineal derivatives of the mesoderm, are generated embryonically and proliferate postembryonically (*Bate et al., 1991*; *Fernandes et al.,*

*For correspondence: vijay@ncbs.res.in

**eLife digest** Muscle tissues must grow and change to accommodate the needs of an animal at various stages in its life. For example, fruit flies begin life as larvae and their muscles must help them move their soft bodies. Later, when the flies mature into adults, the muscles must provide power for flight and support for the insects' external skeletons.

Like other animal tissues, muscles develop from non-specialized stem cells which at first have the potential to become almost any cell type, but later change to become more specialized. Studies of fruit flies, in particular, have yielded insights on how pools of stem cell are created and regulated. Fruit flies are small and easier to study than larger organisms, and as a result, scientists have learned a lot about their genetics and cell biology. Gunage et al. have now identified the stem cell pools that develop into flight muscle tissue, and found that these cells were set aside for the muscles when the fruit fly embryo was still developing.

Fruit flies have large forewings that power flight, and small modified hindwings (called halteres) that help the insect to balance when flying. Gunage et al. reveal that a small, but similar, number of cells are set aside to make both both the tiny muscles that will move the halteres and the much larger flight muscles that move the forewings. However, the cells that contribute to the flight muscles divide to give far more muscle progenitor cells than their haltere counterparts, and make a couple of thousand cells that eventually fuse to form muscle fibers.

Gunage et al. looked at how the flight muscle progenitors multiplied by genetically engineering some of the stem cells in fruit fly larvae so that when each cell divided, its two daughter cells would fluoresce with different colors. One daughter cell would glow green and the other glow red. Gunage et al. found that at first the cells multiply equally, with half the new cells coming from a 'red' stem cell and the other half from a 'green' cell—meaning that the number of cells increases exponentially. Later, the balance shifted so that either more red cells than green cells were produced, or vice versa. This results in a 'linear' increase in number of muscle progenitor cells. Furthermore, Gunage et al. identified the proteins that orchestrate the switch from equal to unequal multiplying of these cells at the different times points in the fruit flies' development.

The next challenge is to see if these stem cells that form the muscles are also available for repair of mature muscle tissue after it is damaged. If this is so, these stems cells might perform a similar function to muscle satellite cells, which are found in the mature muscles of mammals and other vertebrates.

*1991*; *Roy and VijayRaghavan, 1999*). Little is known about the cellular and molecular mechanisms by which the AMPs proliferate and to give rise to the large number of cells which are needed to contribute to the massive adult flight muscles. During late embryogenesis the AMPs required for the formation of flight muscles are set aside in the mesothoracic segment (T2) and those required for haltere muscle development in the metathoracic segment (T3) (*Sudarsan et al., 2001*; *Roy et al., 1997*). The numbers of AMPs at this early stage in T2 and T3 are same but the AMPs in T2 proliferate profusely while those in T3 far less. Studies on the 'four-winged-fly' have clearly shown the key role played by the wing-disc ectoderm in regulating myoblast proliferation (*Fernandes et al., 1994*; *Dutta et al., 2004*; *Roy and VijayRaghavan 1997*). Yet, the mechanisms that regulate the amplification of muscle precursors to generate large 'pools of myoblasts', a feature common to adult muscles in the fly as well as to vertebrate skeletal muscles, (*Sudarsan et al., 2001*) have not been studied in the fly or indeed other systems.

In this report, we use clonal MARCM (*Yu et al., 2009*) techniques to study the proliferative activity of AMPs during postembryonic development. We focus on the AMPs associated with the wing imaginal disc in the second thoracic segment, which give rise to the large indirect flight muscles. We show that an initial amplification of the number of these AMPs occur through symmetric divisions and is followed by a switch to asymmetric divisions, in which the AMPs self-renew and generate postmitotic myoblasts required for the formation of adult myofibers. The sequential nature of these two division modes results in a change in the arrangement of AMP lineages from an initially monostratified layer adjacent to the wing disc epithelium to a markedly multistratified layer comprising both AMPs and their post mitotic myoblast progeny. While the initial amplification of AMPs through symmetric divisions

is controlled by Notch signaling, the switch to the subsequent asymmetric division mode of AMP division additionally requires Wingless. In both cases the epidermal tissue of the wing imaginal disc acts as a stem cell niche and provides the ligands, Serrate and Wingless, for the two signaling pathways that operate in the AMPs. We identify the AMPs as a novel muscle stem cell population whose proliferation pattern orchestrates the building of the large flight muscles in *Drosophila*.

## Results

### Wing disc associated AMPs proliferate during larval development to reach a population size of 2500

AMPs are lineal descendants of progenitor cells, derived from proneural gene-expressing equivalent groups in the embryonic mesoderm (*Ruiz-Gómez and Bate, 1997*; *Carmena et al., 1998*). In the thoracic segments of the embryo, each of these progenitor cells divides asymmetrically to generate a muscle founder cell involved in embryonic myogenesis and an AMP which has active Notch (N) signaling and continues to express the mesodermal marker Twist (Twi). In contrast to their muscle founder cell siblings, the AMPs do not differentiate in the embryo; rather they are set aside and at the end of embryonic development become associated with the imaginal discs (*Bate et al., 1991*). The Twi-expressing AMPs associated with the wing imaginal discs co-express Vestigial (Vg) in late embryonic stages (*Sudarsan et al., 2001*). In the early embryonic stage (Stage11) Wg induced Dll (Distalless) expression leads to separation of the primordium for different discs and those for wing imaginal discs start expressing Vg and results in defining of the population for wing disc primordium (*Cohen et al., 1993*; *Campos-Ortega, 1997*; *Kubota et al., 2000*). To determine the number of AMPs present in the thoracic segments at the end of embryogenesis, we identified these cells using *Twi*-GAL4 > UAS-mCD8::GFP.

In each thoracic segment a total of approximately 10 cells were Twi-positive and dorsally located indicating that these correspond to the dorsal AMPs (*Figure 1A*) (*Williams et al., 1991*; Creig and Akam et al., 1993). To further characterize the extent of post-embryonic proliferation of the flight-muscle AMPs, we determined the number of Twi-positive cells associated with the T2 wing disc at different larval stages. Immediately after larval hatching (24 hr AEL) a total of 10 (±2) Twi-positive cells were found on the wing disc implying that these had not yet initiated proliferative divisions (*Figure 1B*). At the late second instar stage (72 hr AEL) the number of Twi-positive cells associated with the wing disc had increased to approximately 250 (±15) and at the late third instar stage (144 hr AEL) approximately 2500 (±90) Twi-positive cells were found on the wing disc indicative of intense proliferative activity (*Figure 1C–E*). We conclude that the 10 AMPs present on the wing disc at the end of embryogenesis and the beginning of larval development initiate a proliferation process during subsequent larval stages which results in some 2500 Twi-positive cells at the end of larval development (*Figure 1E,F,G*). These findings suggest that each embryonically born AMP, on the average, would give rise to a lineage of approximately 250 cells during larval development. We next investigated the cellular mechanisms that make this remarkable proliferation possible.

### AMP proliferation involves initial symmetric and subsequent asymmetric clonal amplification

To determine the mechanisms of AMP proliferation we used twin-spot MARCM, a genetic labeling technique which makes it possible to trace cell lineage and determine proliferation patterns by independently labeling the two paired sister cell siblings derived from a given cell division (*Yu et al., 2009*; *O'Brien et al., 2011*). In twin-spot MARCM, heat shock induces a timed mitotic recombination event in the precursor cell which results in differential labeling of its two daughter cell clones with heritable expression of green fluorescent protein (GFP) in one cell and red fluorescent protein (RFP) in the other. Symmetric cell divisions will result in daughter cell clones of RFP and GFP of equal number of cells; asymmetric clonal amplification should result in daughter cell clones of GFP and RFP of unequal size.

Twin-spot MARCM clones were induced in proliferating AMP lineages during larval development and recovered in the third instar. Clones were induced using the mesoderm-specific *Dmef2*-Gal4 driver (*Ranganayakulu et al., 1998*). Wing discs containing labeled clones were co-labeled for Twi-immunoreactivity. Twin-spot MARCM clones induced between the early first instar and the late second instar invariably resulted in two relatively large, differentially labeled daughter cell clones

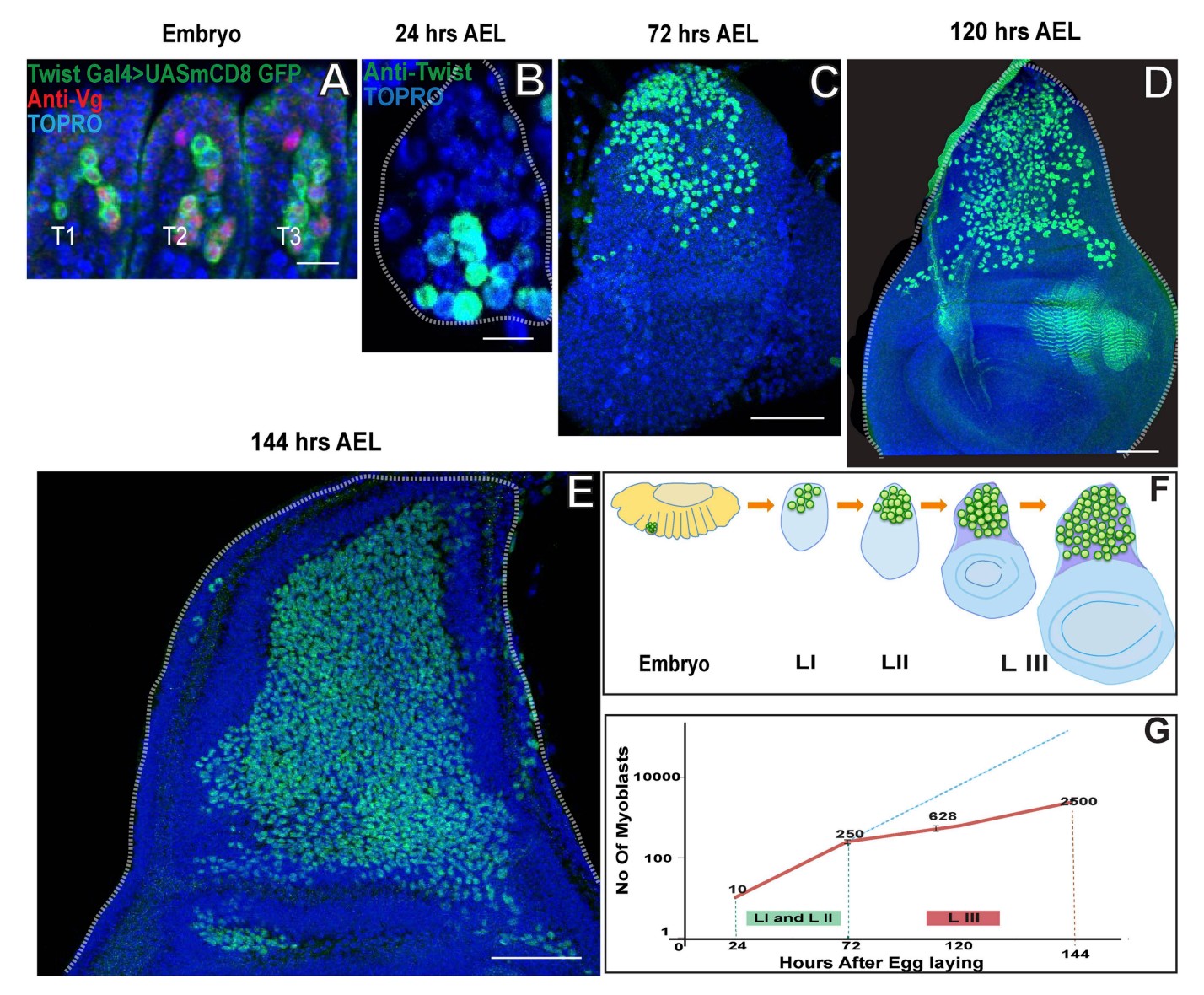

**Figure 1**. Wing disc associated AMPs proliferate during larval development to reach a population size of 2500. (**A**) Stage 17 embryo (~22 hr After Egg Laying (AEL) showing cluster of 10 AMPs in thoracic segment (T2) marked with *Twist* Gal4 > UAS mCD8GFP, Vg (anti-Vestigial, red) and TO-PRO3 (**A** nuclear stain, blue), Similar numbers of Twi positive cells are seen in each segment. n = 5 Scale bar, 10 μm. (**B–E**) Wing imaginal discs from early first (~24 hr AEL) n = 5. Scale bar, 10 μm, late second instar (~72 hr AEL) n = 10 and third instar stage (~120 hr AEL, n = 10 and ~144 hr AEL, n = 10) stained for Twi (anti-Twist, green) and TO-PRO3 (A nuclear stain) showing increase in the number of AMPs during the larval instars. Scale bar, 50 μm. (**F**) Schematic showing AMPs, marked in green color, in T2 region of stage 17 embryo and subsequently in the presumptive notum of the first instar, second instar and late third instar wing imaginal disc. (**G**) A sharp increase is seen in the number of AMPs in first (I) and second (II) instars (Till 72 hr AEL) (Depicted as red line). After 72 hr AEL (Early third instar) till the end of third instar (144 hr AEL), the rate of increase of the AMP population is less sharp. The dotted blue line depicts the extrapolation of the early rate of growth. The graph depicts the average number of cells and the bar represents the standard error. For first instar (24 hr) n = 5, late second instar (72 hr) n = 10, mid third instar (120 hr) n = 10 and late third instar (144 hr) n = 10.

that were similar in size and contained a similar number of cells (*Figure 2A–C*). This indicates that mitotically active cells in the AMP lineages divide symmetrically during early larval development (30 hr–72 hr AEL). In contrast, twin-spot MARCM clones induced during larval development from the early third instar (~75 hr AEL) onward always resulted in two differentially labeled daughter cell clones of different number of cells in each clone (*Figure 2D–F*). Notably, one of the two daughter

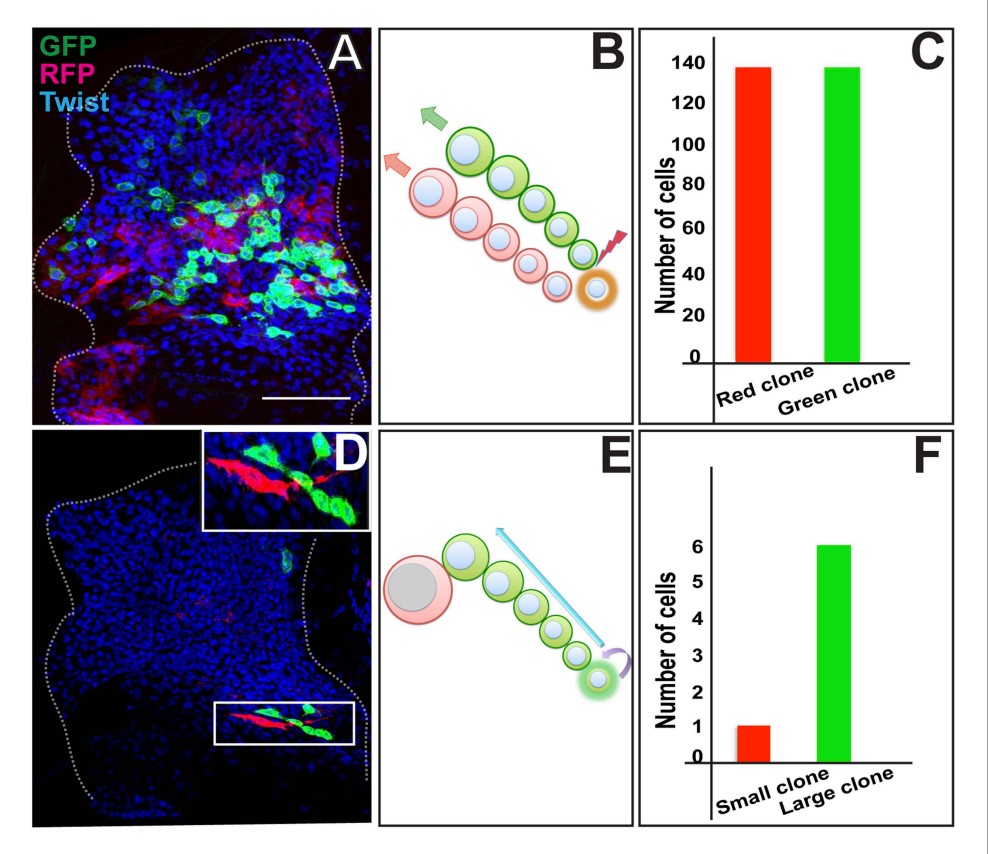

**Figure 2**. AMP proliferation involves initial symmetric and subsequent asymmetric clonal amplification. (**A**) Third instar wing imaginal disc (~144 hr AEL) showing a Twin-spot MARCM clone induced by a single 15 m heat shock at 37°C in the early first instar (~24 hr AEL). The GFP and RFP twin spots are of the same size. Anti-Twist marks all the descendants of AMP lineages. (**B**) Schematic depiction of clonal generation of symmetric clone from the AMP from the early first instar till late second instar stage. N = 10 early clones were examined all with a similar result showing the twin spots of the same size. (**C**) Quantitation of the clone showing exactly same number (136) of cells in both red and green clone of twin-spot. (**D**) Asymmetric clone of AMP lineage recovered from a single 15 m heat shock at 37°C clonal induction in early third instar (~75 hr AEL). Clone shows a single large red cell and six small green cells (zoomed image showed in top most right corner). Scale bar 50 µm. (**E**) Schematic depiction of clonal generation of asymmetric clone from the AMP in the early third instar onwards. N = 12 late clones were examined all with a similar result. (**F**) Quantitation of clone showing one red and six green cells from the twin-spot marking experiment.

cell clones invariably comprised one large labeled cell with elongated morphology, whereas the other daughter cell clone contained up to 10 (±3) smaller cells. This indicates that mitotically active cells in the AMP lineages have differential clonal potential and divide during the third larval instar stage (72 hr–144 hr AEL), with each stem cell giving rise to one stem cell through self-renewal as well as a sibling cell which could differentiate further.

Taken together, these findings indicate that proliferating cells in the AMP lineage divide in a symmetrical mode during early development such that 10 AMP progenitors in the early first larval instar stage give rise to a total of approximately 250 cells at the end of the second larval instar stage. In the third larval instar stage, these myogenic progenitor cells divide to produces up to 10 progeny to generate a total of approximately 2500 myogenic cells. This division is different from the earlier mode, as the division outcome is asymmetric as the majority of the clone is essentially due to proliferation of the one of the cell. Thus, by transiting through a sequence of symmetric cell divisions followed by asymmetric cell divisions, each embryonically generated AMP can generate a lineage of approximately 250 myogenic progeny during larval development.

## Proliferating AMPs are located in a monolayer adjacent to the wing disc epithelium

During the first and second larval instars, AMPs labeled with the Twi are located in a single monostratified layer adjacent to, and in contact with, the wing disc epithelium (*Figure 3A,B,B'*). From the early third larval instar onward, labeled cells become organized in multistratified 2-3 cell layers on the wing disc (*Figure 3C,C'*; *Video 1*). To determine if mitotic activity takes place in specific layers, cells were co-labeled for phospo-histone-3 (PH3) immunoreactivity and *Dmef2*-Gal4 was used to drive UAS mCD8::GFP. Throughout larval development, PH3 positive cells were only observed in the cell layer immediately adjacent to the wing disc epithelium. Notably, even in the multistratified organization manifested in the third larval instar, PH3 positive cells were seen only in cells located adjacent to the epithelium and not in cells located in more distal layers (*Figure 3D–D″*). This implies that almost all mitotically active cells are located proximal to the disc epithelium, while the cells in the more distal layers are likely to be postmitotic. Taken together with the findings on cell division modes in the AMP lineages, these results also suggest that both symmetric and asymmetric cell divisions only take place in cells located adjacent to the wing disc epithelium.

Since twin-spot MARCM experiments indicate that a transition from a symmetrical to an asymmetrical mode of cell division occurs between the second and third larval instar stage, we focused on the mitotically active cells in the layer adjacent to the wing disc epithelium and examined the orientation of the axis of division at these two stages. Centrosome position and, hence, the axis of cell division were revealed by a GFP-tagged peri-centrosomal (CNN-GFP) label (*Megraw et al., 2002*) and *Dmef2*-Gal4 was used to drive UAS-CNN-GFP expression. Throughout the second instar stage, the axis of cell division in the mitotically active cells is oriented parallel to the surface of the wing disc epithelium such that both daughter cells remain in contact with the epithelium (*Figure 3E–H'*). In contrast, from early third instar stage onward, the axis of cell division shows an oblique to orthogonal orientation relative to the disc surface (*Figure 3I–L'*). Consequently, only one of the two daughter cells remains in contact with the disc epithelium while the other daughter cell loses epithelial contact and, hence, might contribute to the population of cells located in layers distal to the disc epithelium.

## Proliferating AMPs generate clones of postmitotic myoblasts localized in distal layers

Given that proliferative mitotic activity in the AMP lineages only takes place in the cells located next to the epithelium we performed experiments to confirm that the cells in the more distal layers are indeed their postmitotic progeny. For this, we first performed EdU (5-Ethynyl-2'-deoxyuridine) incorporation experiments to visualize cells actively engaged in DNA replication as well as their progeny. EdU incorporation was carried out in early third instar larva (in a 5 hr pulse) and visualized in AMP cells on wing discs co-labeled with for Twi immunoreactivity. When cells were examined directly after EdU incorporation (pulse, no chase), labeling was only found in cells adjacent to the disc epithelium (*Figure 4A,B*). This result confirms the finding that mitotically active cells are located next to the epithelium. If the incorporation of EdU was followed by 48 hr without EdU (pulse, chase) and the cells then examined, labeling was seen in all layers. However, in this case, the majority of labeled cells were in the most distal layer, less labeled cells were in an intermediate layer and the lowest number of labeled cells was found in the layer adjacent to the disc (*Figure 4C,D*). This indicates that the pulse of EdU labeling incorporated into the (replicating) cells adjacent to the disc epithelium was now manifest in the (non-replicating) cells located in more distal layers. This, in turn, implies that the cells in the more distal layers are the postmitotic progeny of the proliferating cells that remain in contact with the disc epidermis.

To further investigate the clonal nature of this proliferative process in the AMP lineages, we next carried out MARCM labeling experiments using the *Dmef2*-Gal4 driver. MARCM clones induced in the second larval stage and recovered in the third larval stage typically comprised 100–200 labeled cells arranged in a loose cluster and extending from the layer next to the disc epithelium to the most distal layer of cells (*Figure 4E*). Interestingly, within each clone the labeled cells in the distal layer were larger in size and had a more elongated morphology than those located in the innermost layer next to the disc (*Figure 4G–N*; *Video 2*). To determine if these clones comprised mitotically active cells, we combined MARCM labeling with PH3 immunolabeling. In all cases in which co-labeling was seen, only one cell within the clone was PH3-positive and this cell was always located adjacent to the disc epithelium (*Figure 4F,F'*; *Video 3*).

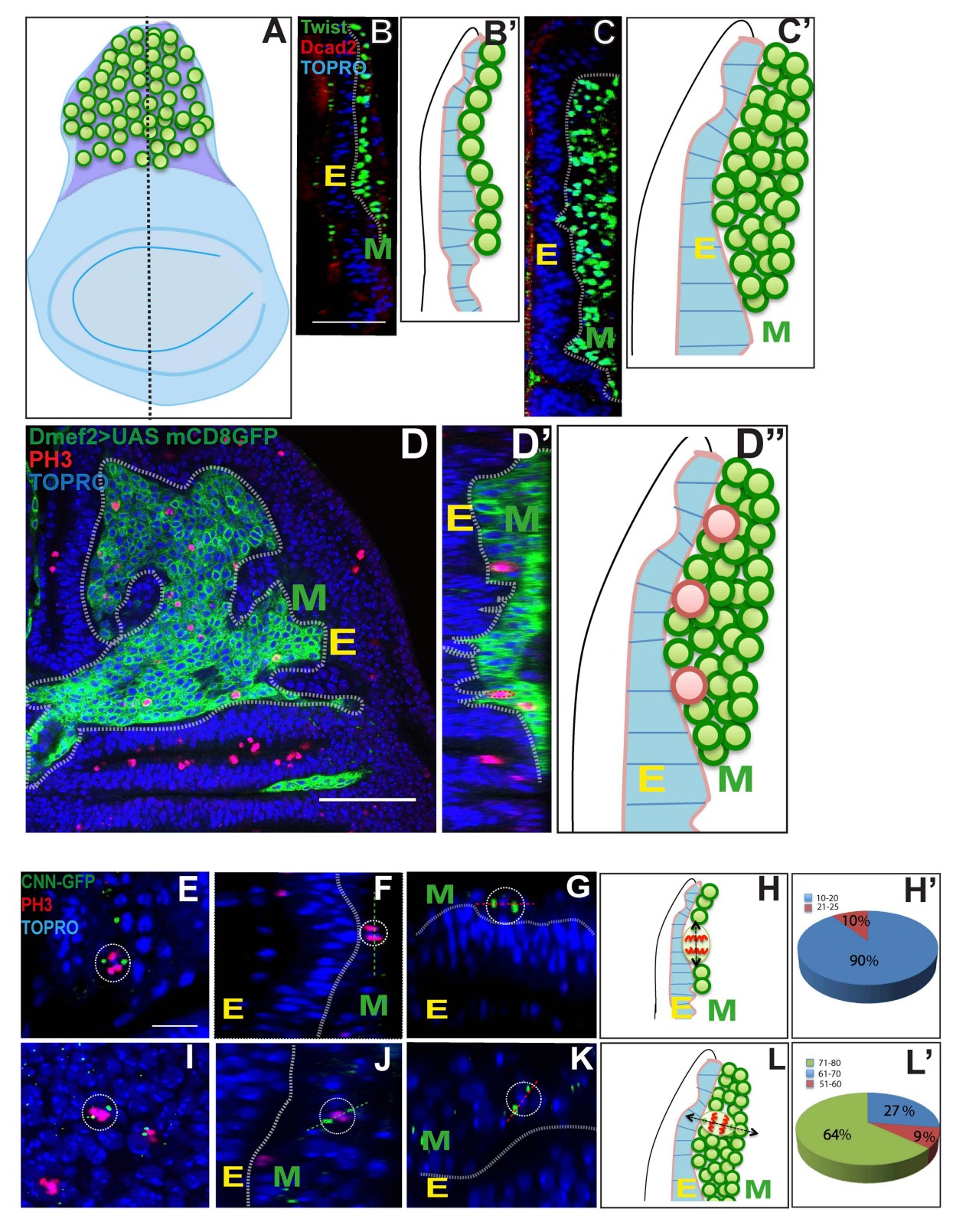

**Figure 3**. Proliferating AMPs are located in a monolayer adjacent to the wing disc epithelium. (**A**–**B'**) Optical section of late second instar wing imaginal disc (~72 hr AEL, schematic in **A** with AMPs in green), showing monostratified (single layer) arrangement of AMPs (marked in green and denoted as M for mesoderm, **B**) stained for Twi (anti-Twist, green), Dcad2 (anti-Dcad2, red) (Disc-epithelium [E]) and TO-PRO3 (a nuclear stain, blue). n = 10, Schematic in **B'**.
*Figure 3. Continued on next page*

*Figure 3. Continued*

(**C–C'**) Late third instar wing disc showing multistratified (3–4 layers) of AMPs in presumptive notum region. n = 20 Scale bar 50 μm. AMPs are stained for Twi (anti-Twi, green), epithelium (anti-Dcad2, red) and TOPRO (Nuclear stain, blue) (**D–D''**) A late third instar wing imaginal disc showing multistratified arrangement of AMP using *Dmef2*-Gal4 > UASmCD8GFP (anti-GFP, green) and co-stained for mitotic marker PH-3 (anti-phosphohistone, red) and TO-PRO3 (Nuclear stain, blue). **C'** shows a schematic representation. **D'** shows an optical section of image **D** showing active mitotic divisions (red dotted circles) exclusively in the layer most-proximal to the disc epithelium (marked as E). n = 20. **D''** represents this schematically. (**E, F** and **G**) Late second instar wing discs (~70 hr AEL) marked for centrosomin-GFP (CNN-GFP, a pericentrosomal marker, green) using *Dmef2*-Gal4, PH3 (anti-phosphohistone, red) and TO-PRO3 (Nuclear stain, blue). In each panel, E is the Epidermis and M the AMPs. (**H–H'**) Schematic (**H**) of a late second instar disc (as shown in **F**) showing active mitotic division (Black arrow) parallel to the disc epithelium. Pie-chart representation (**H'**) (Blue: 10°–20° with 0° being parallel to the epidermis. n = 6 preps. (**I, J** and **K**) Third instar wing imaginal disc (~140 hr AEL) showing a more orthogonal orientation in AMP marked using centrosomin-GFP (CNN-GFP) (a centrosome marker, green) using *Dmef2*-Gal4, PH3 (anti-phosphohistone, red) and TO-PRO3 (Nuclear stain, blue). n = 10. Scale bar, 10 μm. (**L–L'**) A schematic representation (**L**) and pie-chart representation (**L'**) of late third instar disc showing mitotic division axis (black arrow, in **L**) of AMP (green) oblique to disc surface (blue). The angle of division at this stage is in the range of 50–80° as represented in **L'**.

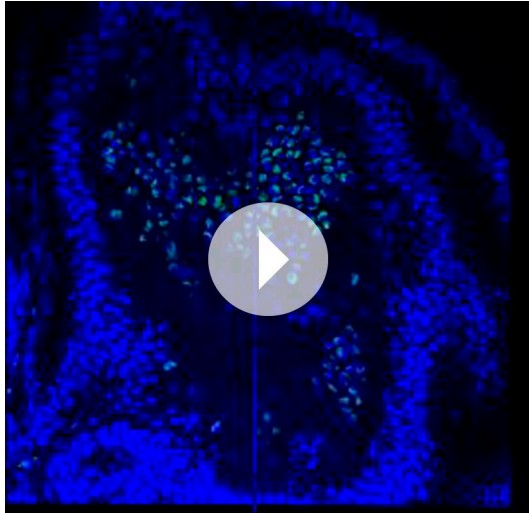

**Video 1**. Showing multilayered arrangement of myoblasts (related to *Figure 3*). 3D reconstruction of third instar wing disc showing AMP lineage stained with anti-Twi (green) and all disc nuclei stained using TOPRO (blue) showing stacked arrangement.

Taken together, these findings indicate that proliferation in the AMP lineages occurs in two phases during larval development. In early larval development (first and second instar), AMPs undergo a marked amplification in cell number through symmetric divisions. These divisions are oriented parallel to the wing disc epithelium and hence the AMPs form a monolayer of proliferating cells that remain in contact with the disc epithelium. In late larval development (third instar) AMPs switch to an asymmetric mode of proliferative cell division in which they self-renew and generate postmitotic daughter cells. These asymmetric divisions are oriented obliquely to the wing disc surface such that the self-renewing daughter cells remain in contact with the disc epithelium while the postmitotic daughters become located in the more distal layer of non-replicating myoblasts.

## Notch signaling is required for proliferative activity of AMP lineages

Previous work has shown a requirement for Notch (N) in adult flight muscle development; in N mutants a depletion of wing disc myoblasts is seen (***Anant et al., 1998***). This suggests that N signaling might be involved in the proliferative activity of AMP lineages. To investigate this further, we first determined if N is expressed in the (Twi-positive) cells on the larval wing disc using an anti-NICD (Notch Intracellular Domain) antibody. Immunocytochemical analysis shows that all of the Twi-positive cells are co-immunolabeled by the anti-NICD antibody (***Figure 5A–C,F–H***). This co-immunolabeling was seen in all larval instar stages and in all myogenic cell layers associated with the wing disc.

To determine the role of N in these Twi-positive cells, we used the *Dmef2*-Gal4 driver together with UAS-*N* RNAi to down-regulate N in the AMPs and then assayed mitotic activity using PH3 immunoreactivity in late third instar stage. (Gal80^ts was used to limit N-RNAi to the second and third larval instar to avoid lethality.) A significant reduction in the number of mitotically active cells was observed; in the third instar stage only half the number of PH3-positive cells were seen in knockdown vs control experiments (***Figure 5D***). Similar findings were obtained in experiments in which a dominant negative form of N was expressed using the *Dmef2*-Gal4 driver (data not shown). Both the number and the layered organization of the ensemble of Twi-positive cells on the wing disc were reduced in these knockdown experiments (***Figure 5I–K***). Conversely, overexpression of an activated form of N in the AMP using *Dmef2*-Gal 4 to drive UAS-*NICD* in second and third larval instar stages revealed a marked increase in mitotically active cells as assayed in late third instar stage. In these experiments the number

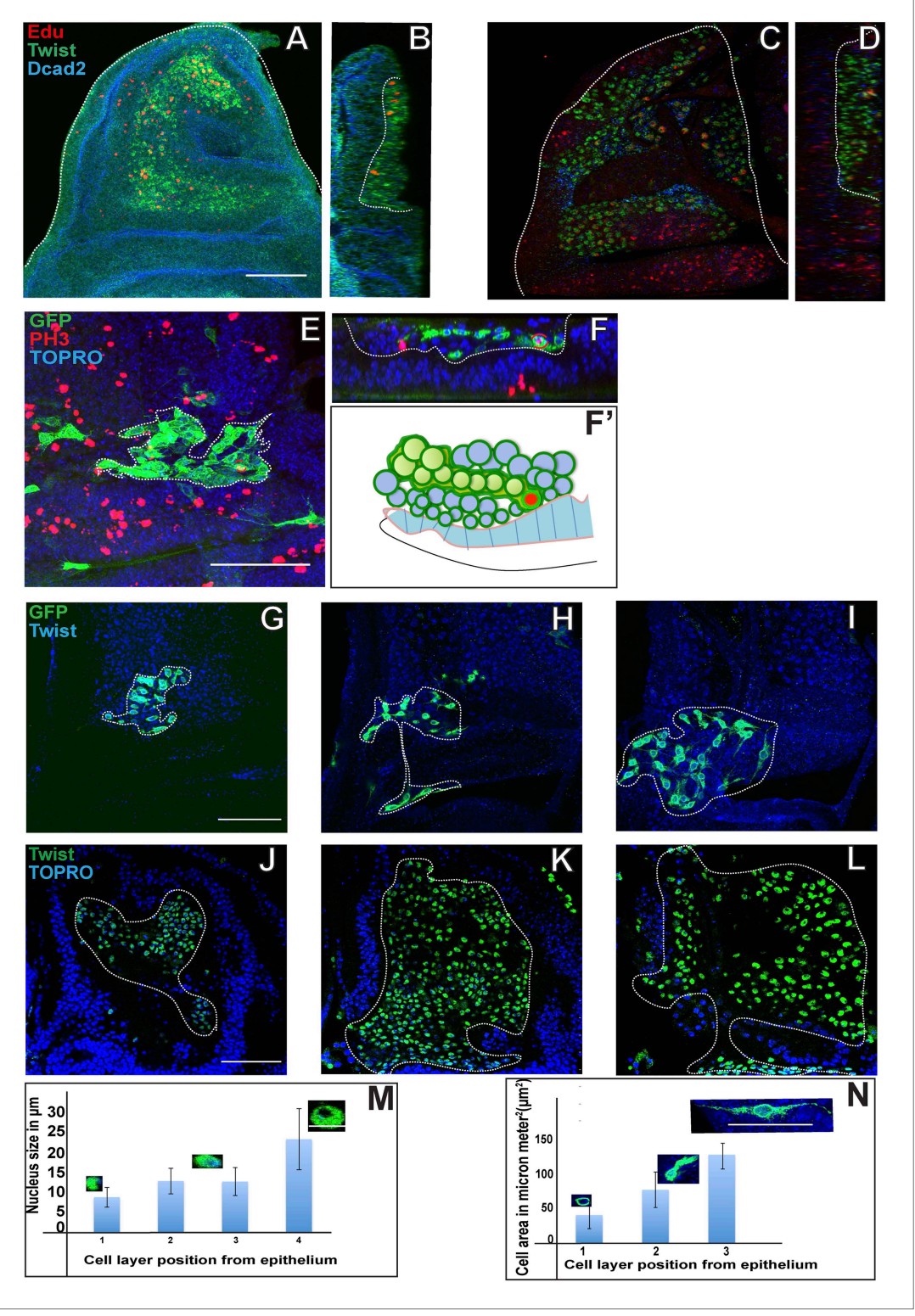

**Figure 4**. Proliferating AMPs generate clones of post mitotic myoblasts localized in distal layers. (**A**) Optical section of third instar wing disc dissected after 5 hr Edu, a thymidine analogue (5-ethynyl-2′-deoxyuridine) pulse and stained for Edu (red), Twi (anti-Twist, green) and Dcad2 (anti-Dcad2, blue) revealing presence of Edu in some AMP lineage. n = 10. Scale bar, 50 μm. (**B**) Sagittal section of presumptive notum region of disc (shown in **A**) showing Edu labeling of AMP lineage only next to disc epithelium surface. (**C**) Third instar disc dissected after 5 hr Edu pulse and 48 hr

*Figure 4. Continued*

chase in pulse free media and stained for Edu (red), Twi (anti-Twist, green) and Dcad2 (anti-Dcad2, blue) showing labeling in the most distal layer of AMP lineages. n = 12. (**D**) Sagittal section of **C** showing labeling in maximum labeling in distal layers and minimum in the layer next to epithelium. (**E**) A MARCM clone generated in second instar stained for GFP (anti-GFP, green) and PH3 (anti-Phosphohistone, red), TO-PRO3 (blue). The clone shows a single PH3 positive cell (shown in red bracket) in the cluster of other clonal progenies. n = 10. (**F–F'**) Optical section (**F**) and schematic (**F'**) of figure E, showing the presence of PH3 positive AMP (red bracket) next to the disc epithelium (dotted line). (**G–I**) The MARCM clone showing clonal progenies marked with GFP in the proximal most (**G**), middle (**H**) and distal most (**I**) layers with reference to the disc epithelium. The graph (**N**) shows increase in the cell size from **G** to **J**. n = 12 Scale bar 50 µm. (**K–N**) The optical sections of late third instar disc stained for Twi (anti-Twist, green) and TO-PRO3 (blue) and the quantitation (**M**) showing trend of increase nuclear size in the layers distal most in reference to epithelium. n = 10. Scale bar 50 µm.

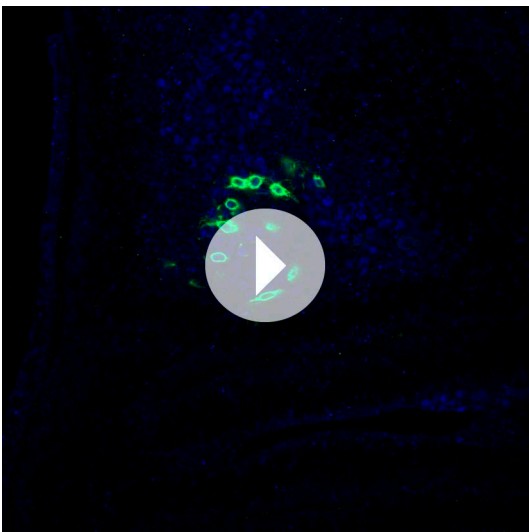

**Video 2**. MARCM clone showing variation of cell size (related to *Figure 4*). 3D reconstruction of AMP lineage marked by membrane tethered GFP (mCD8::GFP) revealing size differences in the myoblasts at different distances from disc epithelium. All nuclei marked by TOPRO (blue).

of PH3-positive cells in the overexpression experiments was approximately twice as large as in controls (*Figure 5E*). Correspondingly, both the number and the layered organization of the Twi-positive cells on the disc were increased in these overexpression experiments (*Figure 5L–N*). Interestingly, in contrast to wild type controls, mitotically active cells were occasionally seen in more distal cell layers in N overexpression experiments (data not shown).

These findings indicate that proliferative activity in the AMP lineages during larval development is dependent on N signaling. Since mitotic activity in the wild type is only seen in the layer of cells adjacent to the wing disc this further implies that N signaling acts to effect proliferation in the AMPs in this layer.

## Serrate located in the wing disc epithelium is necessary for AMP proliferation

The N ligand Serrate (Ser) has been shown to be present in the dorsal (presumptive notum) region of the wing disc in which the AMP lineages develop (*Speicher et al., 1994*). This suggests that Ser might be the ligand for the N signaling required in AMP proliferation. To investigate this, we first determined if Ser expression in the disc epithelium occurs in close proximity to AMPs in the adjacent myogenic cell layer. Immunocytochemical labeling of the wing disc epithelium (with anti-DCad2) and of the AMP lineages (with anti-Twi) together with a Ser-specific marker (Ser-LacZ) reveals a close apposition of Ser-expressing wing disc cells and the AMP cell layer immediately adjacent to the disc epithelium (*Figure 6A–H'*; *Video 4*). This close apposition was observed in all larval stages.

The proximity of Ser in the wing disc epithelium to mitotically active AMPs suggests that epithelial Ser could be the ligand that activates N signaling in the AMPs. To investigate if wing disc-derived Ser is required for N-mediated proliferative activity of AMPs, we conditionally knocked down Ser specifically in the dorsal wing disc region of second instar stages with *Ap*-Gal4 driving UAS-Ser RNAi using the TARGET system (*McGuire et al., 2003*; *Martín and Morata, 2006*) and assayed mitotic activity based on PH3 immunolabeling in the late third instar stage (The *Ap*-Gal4 Gal4 expression in other regions in the larva has no direct effect on these results when perturbation were performed during mentioned time scale). A marked reduction in the number of mitotically active cells was observed; in the third instar stage less than half the number of PH3-positive cells were seen in knockdown vs control experiments (*Figure 6I*).

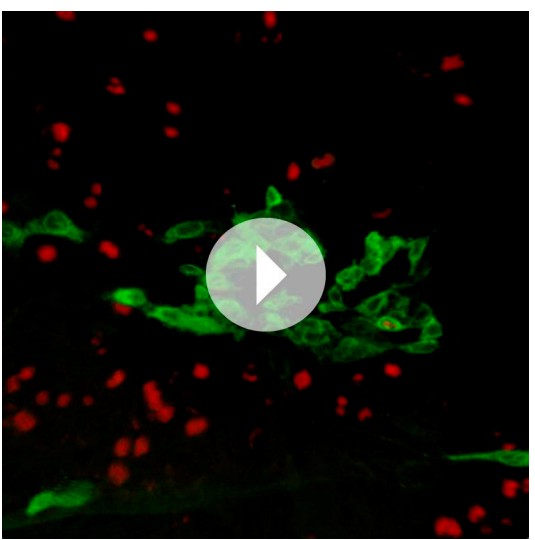

**Video 3**. MARCM clone showing PH3 association with respect to layers (related to **Figure 4**). Single cell in a clone (anti-GFP, green) showing active mitotic division (anti-Phospho histone 3, red).

These results indicate that the Ser located in the wing disc epithelium is required for the mitotic activity of cells in the adjacent AMP layer. Given the concomitant requirement of N for mitotic activity of these cells, this implies that epithelial Ser serves as a ligand for the activation of N receptor-mediated signaling in the myogenic AMP lineages. This in turn suggests that the Ser-expressing cells in the wing disc epithelium represents a transient signaling niche for N activation of AMP proliferation.

## Numb expressed in the third instar stage is required for asymmetric cell divisions in AMP lineages

While the activation of N signaling by Ser is required for mitotic activity in AMP lineages, it is unlikely to mediate the switch from symmetric to asymmetric divisions that occurs during late larval development, since expression of N and Ser in these lineages occurs throughout larval development. Numb (Nb) is an intracellular protein, which binds to N and causes degradation of this receptor ultimately leading to down-regulation of N signaling (**Frise et al., 1996**; **Couturier et al., 2012**). Previous work on mitotic activity of muscle progenitors during embryonic development has shown that Nb acts in asymmetric cell division by down regulating N activity in one of the progenitor's two daughter cells; in this cell downregulation of N activity results in specification of muscle cell fate, while continued N activity in the other daughter cell maintains progenitor cell fate (**Ruiz-Gómez et al., 1997**; **Carmena et al., 1998**).

To determine if Nb is involved in asymmetric cell divisions of AMPs we first carried out immunolabeling studies of Nb expression during different larval instar stages. These experiments show that Nb is expressed in AMP lineages during the third instar stage but not in the two preceding larval stages (data not shown). Throughout the third instar stage, Nb is expressed in all of the myogenic cells on the wing disc, however expression is strongest in the cells of the more distal layers and relatively weak in the cells adjacent to the disc epithelium; this contrasts with N expression which is more uniformly distributed in all of these cell layers (**Figure 7A–D**). The fact that Nb is not expressed in AMP lineages during their symmetric division phase, but is present in these cells during their asymmetric division phase suggests that Nb might be required for the transition to asymmetric cell divisions in AMP lineages.

To investigate this, we carried out twin-spot MARCM labeling experiments and used *Dmef2*-Gal4 driving UAS-*Nb* RNAi to knockdown Nb in the proliferating AMP lineages. Clones were induced in the early third instar stage and recovered in the late third instar stage. These twin-spot MARCM clones invariably consisted of two relatively large, differentially (RFP and GFP) labeled daughter cell clones that were similar in size and cell number indicative of symmetric divisions (**Figure 7F–H**). These clones were similar to the corresponding twin-spot clones obtained in wild type first and second instar stages but strikingly different from those obtained in the wild type third instar (see **Figure 2**). This indicates that in the absence of Nb, cell divisions in AMP lineages are symmetric and not asymmetric in the third instar stage, implying that Nb expression is necessary for the transition from the symmetric to the asymmetric division mode manifest in the third instar.

Given that Nb inhibits N signaling, upregulation of N should have similar phenotypic effects as downregulation of Nb. To investigate this, we repeated the twin-spot MARCM labeling experiments but used *Dmef2*-Gal4 driving UAS-*NICD* to constitutively activate N in the proliferating AMP lineages. Clones induced in the early third instar stage and recovered in the late third instar stage again consisted of two relatively large, differentially labeled daughter cell clones that were similar in size and cell number indicating a symmetric division mode (**Figure 7I–K**). Since both downregulation of Nb and constitutive upregulation of N lead to continued symmetric cell divisions in the third instar stage, we postulate that Nb, through downregulation of N, is required for asymmetric proliferative divisions in the AMP lineages.

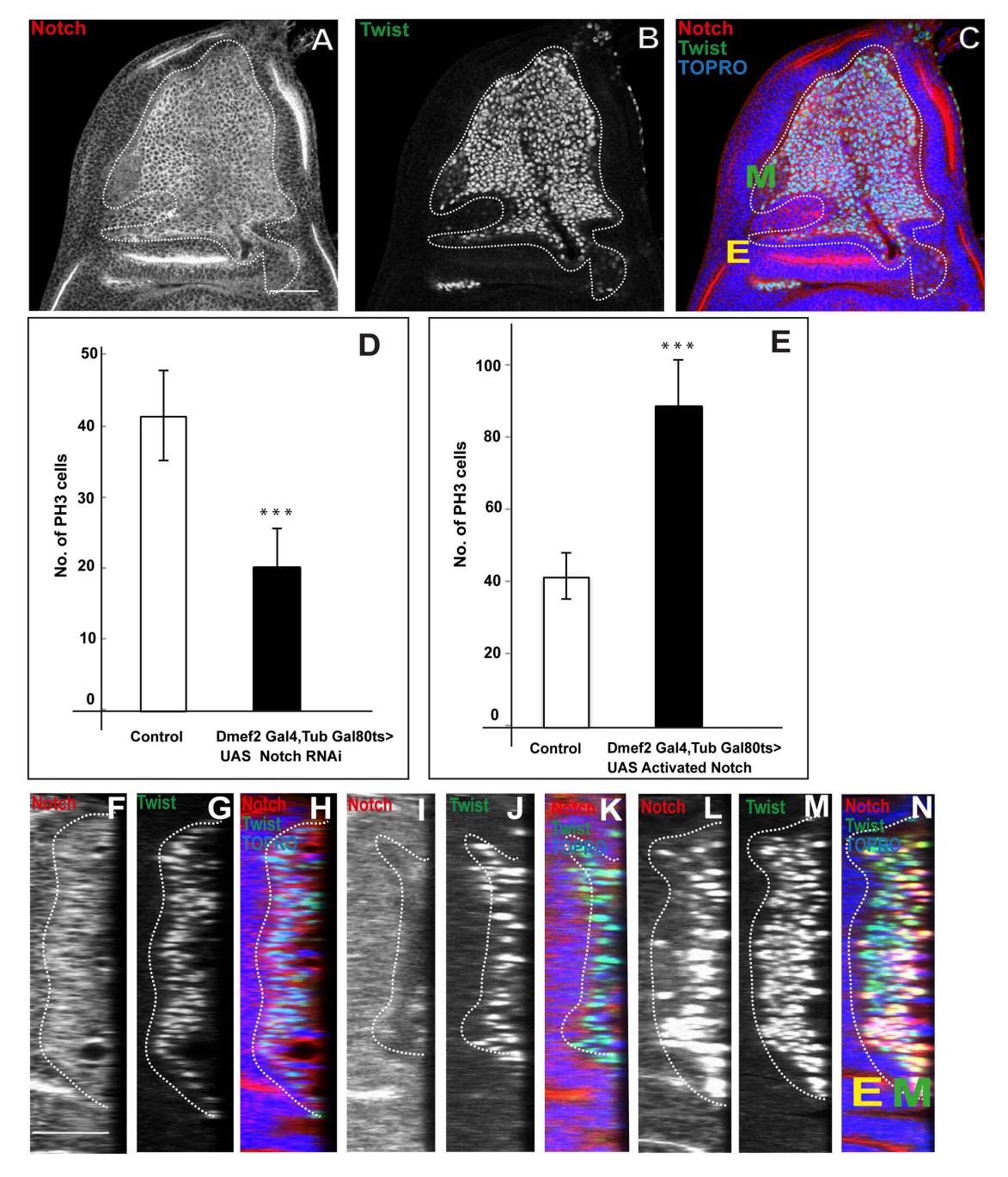

**Figure 5**. Notch signaling is required for proliferative activity of AMP lineages. (**A–C**) Late third instar disc stained for Notch (anti-Notch intracellular C-terminal domain, [NICD, red]), Twi (anti-Twist, green) and TO-PRO3 (blue). In this figure, E denotes disc-epithelium and M is for mesoderm (green). (**D–E**) Quantification of number of PH3 positive AMPs in the Notch donwregulation using *Dmef2*-Gal4, TubGal80ts > UAS *Notch* RNAi (**D**) and Notch up regulation using *Dmef2*-Gal4, TubGal80$^{ts}$ > UAS *NICD* background (**E**). For both experiments Gal80 repression was relieved from early second instar till late third instar by shifting from 18°C to 29°C. All graphs are Mean ± Standard Error (Student's *t* test). n = 12. p-values < 0.001. (**F–N**) Optical section of wing discs stained for Notch (anti-Notch intracellular C-terminal domain, (NICD, red), Twi (anti-Twist, green) and TO-PRO3 (blue) in control (**F–H**) (*Dmef2*-Gal4, TubGal80ts > Canton-S), in *Dmef2*-Gal4, TubGal80ts > UAS *Notch* RNAi (**I–K**), *Dmef2*-Gal4, TubGal80ts > UAS *NICD* (**L–N**). The multistratified layered arrangement of AMP lineages is lost in Notch down regulation (**J**) while in Notch up regulation (**M**) it increases with respect to number of layers, in comparison to control (**G**). Gal80 repression was relieved from early second instar till late third instar by shifting from 18°C to 29°C. n = 20. Scale bar, 50 μm. E denotes disc-epithelium and M is for mesoderm (marked by anti-Twist, green).

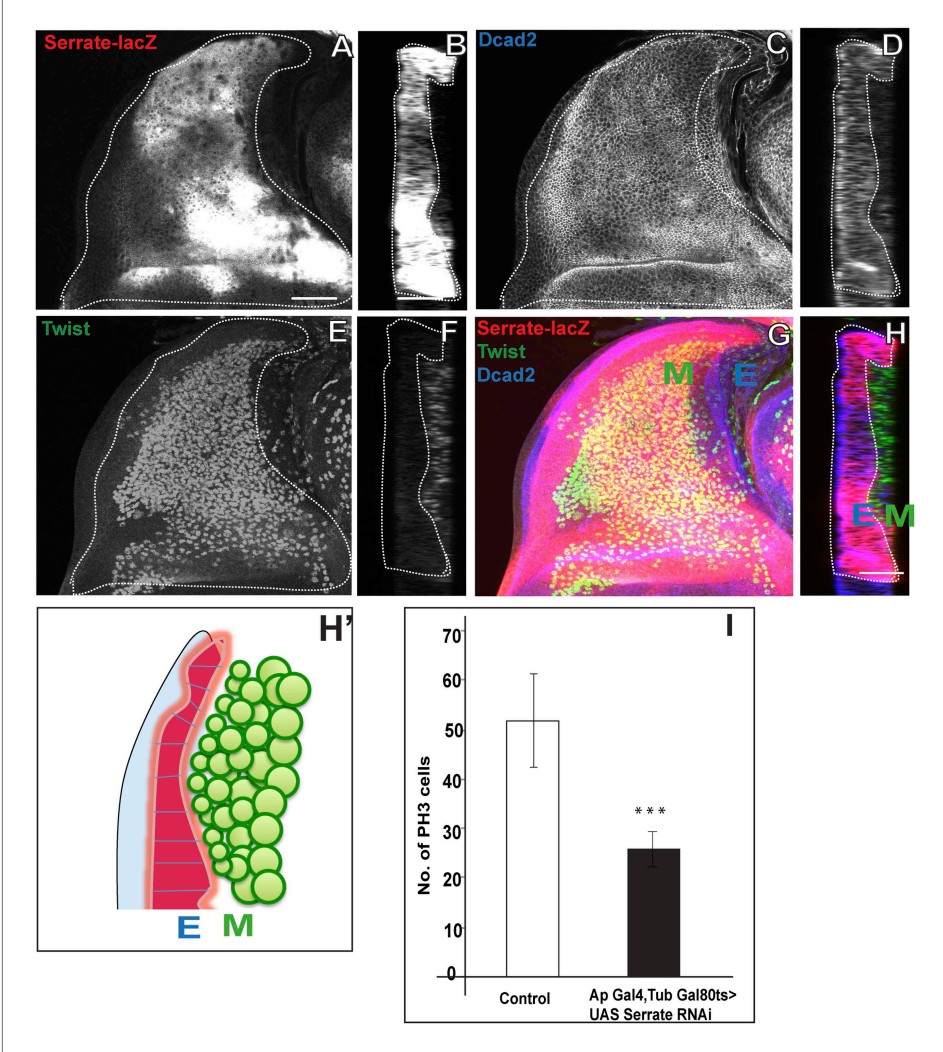

**Figure 6**. Serrate located in the wing disc epithelium is necessary for AMP proliferation. (**A**) Serrate-lacZ (anti-beta Gal), a reporter for Serrate expression, visualized in disc epithelium of the late third instar. (**B**) Saggital section shows Serrate being expressed specifically in the disc epithelium. (**C–D**) Dcad2 expression (anti-Dcad2) marking the disc epithelium. (**E–F**) AMP lineages viewed by using Twi staining (anti-Twist). (**G–H**) The merge shows the expression of Serrate (red) exclusively in the disc epithelium as Dcad2 (blue) and in close proximity of first layer AMP lineages. E-wing disc epithelium, M-AMP lineages. Scale bar 50 μm. **H'**—Schematic depiction of expression patterns of Serrate, Dcad2 and Twist showing only one layer of AMP out of 3–4 (As shown in **F**) is in direct contact with Serrate producing epithelium surface. (**I**) Quantification of number of PH3 positive AMPs in disc epithelium specific Serrate knock down using *Ap* Gal4 driving UAS *Serrate* RNAi. Gal80 repression was relieved from early second instar till late third instar by shifting from 18°C to 29°C. All graphs are Mean ± Standard Error (Student's *t* test). n = 15. p-values < 0.001.

## Wingless signaling from the wing disc epithelium induces Numb expression in third instar AMP lineages

While above mentioned experiments indicate that Nb expression in AMP lineages is important for the transition from symmetric to asymmetric proliferation divisions, they do not identify the signal that controls the onset and maintenance of Nb expression in the third instar stage. Wingless (Wg) is a signaling protein that can act to control developmental patterning and growth (*Sharma and Chopra, 1976*; *Zecca et al., 1996*; *van Amerogen and Nusse, 2009*). During larval development of the wing disc, Wg is expressed in the prospective wing blade region in all instar stages, however, expression in the prospective notum region, in which the developing AMP lineages reside, is only manifest from the

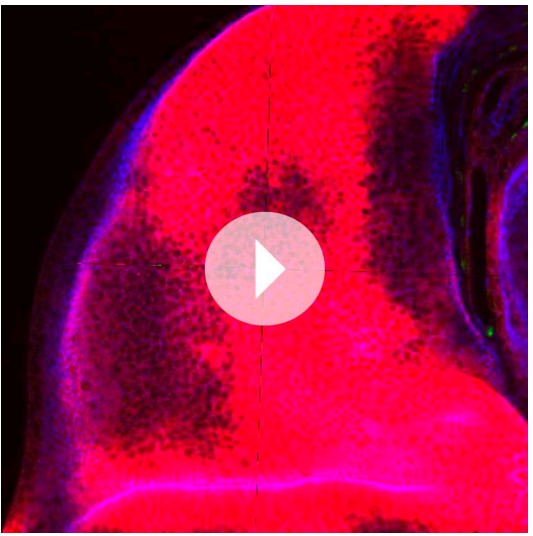

**Video 4**. Serrate staining in epidermis and close association with myoblasts (related to **Figure 6**). Serrate- lacZ (anti-beta-gal, red) showing serrate expression in wing disc epithelium (anti-Dcad 2, blue) and only one layer of myoblasts are in contact with epithelium due to multilayered arrangement.

early third instar onwards (**Phillips and Whittle, 1993**; **Tomoyasu et al., 2000**). Previous work on postembryonic myogenesis has shown that Wg signaling from this region of the larval wing disc epithelium is involved in elaboration and maintenance of myoblast diversity (**Sudarsan et al., 2001**).

To investigate if Wg signaling from the notum region of the disc epithelium might be involved in the induction of Nb expression in third larval instar AMP lineages, we first characterized the spatial expression pattern of Wg in the epithelium relative to the Twi-positive myogenic cells in immunolabeling studies of third larval instar stages. Throughout the third instar stage, notal Wg expression is seen in a stripe of epithelial cells that are closely apposed to the adjacent population of Twi positive cells (**Figure 8A–H**; **Video 5**). To determine if Wg signaling from this spatially restricted stripe of epithelial cells can influence all of the Twi-positive cells in the myogenic layers, we used immunolabeling to visualize beta-catenin activity in wild type vs notal Wg loss-of-function using *Wg(ts)/Wg(Sp-1)* alleles. In these experiments, notal Wg loss-of-function resulted in complete loss of beta-catenin activity in all Twi-positive cells implying that inductive Wg signaling from the notum region of the wing disc affects all myogenic cells in the third larval instar (**Figure 8I–N**). We next carried out similar Wg loss-of-function experiments together with Nb immunolabeling to investigate if inductive Wg signaling is required for onset and maintenance of Nb expression in the third instar myogenic cells. Wg loss-of-function in the wing disc notum resulted in complete loss of Nb expression in all Twi-positive cells (**Figure 8O–V**). We conclude that Wg signaling in the wing disc notum is required for induction and maintenance of Nb expression in the myogenic AMP lineages in the third larval instar stage.

In addition to its effects on Nb expression, notal Wg loss-of-function also results in the reduction of mitotic activity in the AMP lineages. Thus, both the number of PH3 positive cells and the extent of the multilayered organization are reduced in Wg downregulation compared to wild type (**Figure 9A–J**, **Figure 9—figure supplement 1**). To understand the possible direct link of the induction of Wg signaling and change in the orientation of the cell division, we down regulated Wg using DN-TCF with *Dmef2*-Gal4 and probed for division axis using CNN-GFP. The axis of the division in this background is parallel to the epithelium, similar to that seen in the early instars (**Figure 9—figure supplement 2**). Wg, in the presumptive notum is expressed only from the late second instar (**Alexandre et al., 2014**), therefore this result shows that down-regulation of Wg prevents the switch in cell-division orientation from parallel to the epithelium to perpendicular to the epithelium. This suggests that Wg signaling from the wing disc niche might have dual effects on neighboring myogenic cells, namely the induction of Nb expression and the regulation of proliferation.

## Discussion

Early studies in *Calliphora* (**Crossley, 1972**) and elegant clonal analysis and transplantation experiments in *Drosophila* (**Brower et al., 1981**; **Lawrencea, 1982**) established mesodermal cells attached to the wing imaginal disc as the source of muscle progenitors for the large indirect flight muscles of the adult thorax of dipteran insects. These 'adepithelial' cells were subsequently shown to express the mesodermal marker Twist (**Thisse et al., 1988**; **Currie and Bate, 1991**; **Fernandes et al., 1991**). Twist labeling of the AMPs on the late-third instar wing imaginal disc reveal a large number of cells (**Currie and Bate, 1991**; **Fernandes et al., 1991**), which this study identifies as about 2500. The haltere (T3) disc and the wing (T2) disc, start off with a similar numbers of myoblasts in the late embryo. Studies on muscle development in the 'four-winged fly', where the function of the homeotic gene Ultrabithorax

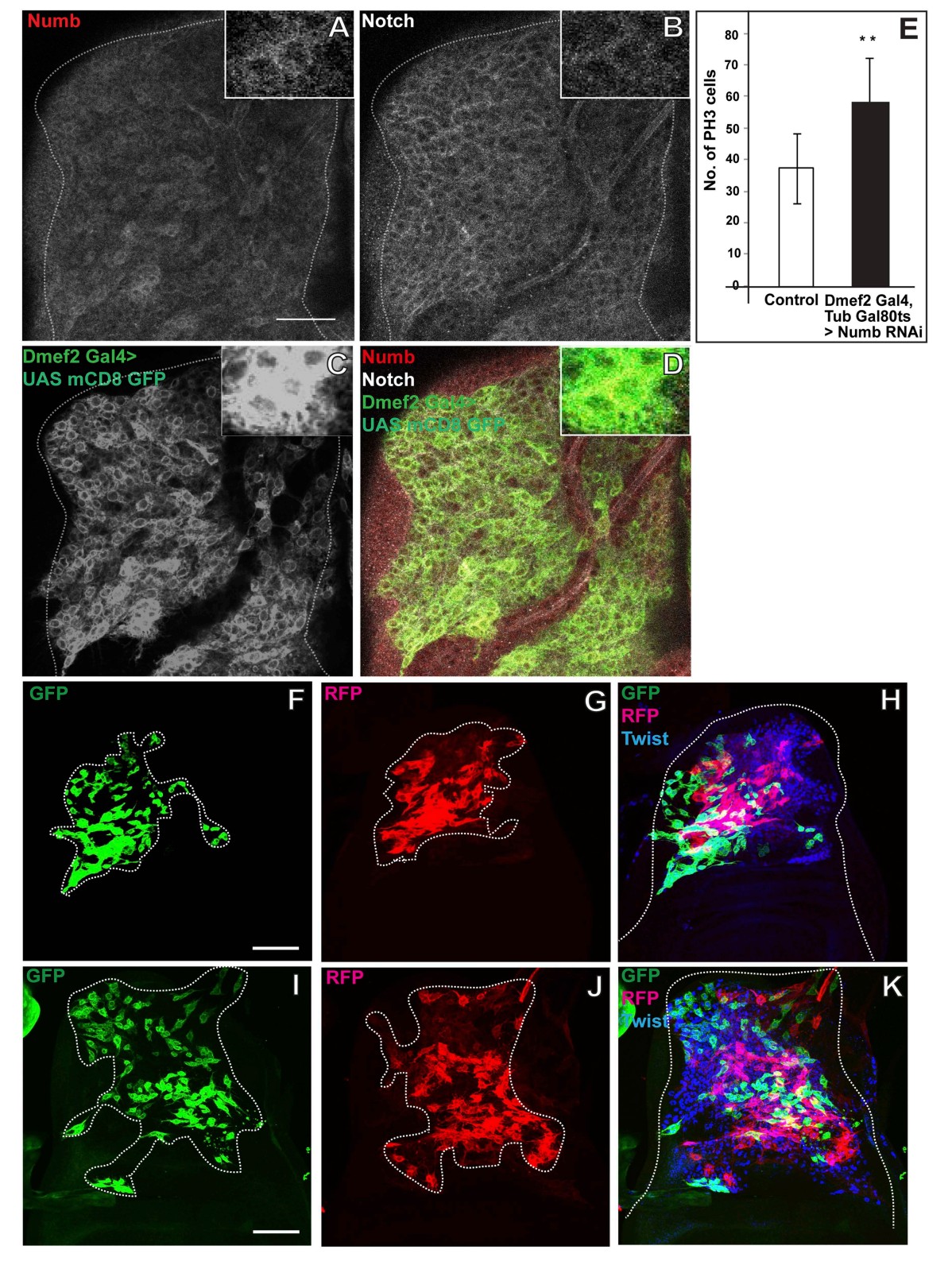

**Figure 7**. Numb expressed in the third instar stage is required for asymmetric cell divisions in AMP lineages. (**A**–**D**) The late third instar disc stained for expression of Numb (anti-Numb, red), Notch (anti-NICD, white) and *Dmef2*-Gal4 > UAS mCD8GFP (anti-GFP, green). N = 6.The expression of Numb can be seen in patches (**A**) in contrast to Notch (**B**) which stains most of the AMP lineages marked by using *Dmef2*-Gal4 > UAS mCD8GFP (**C**). The merge (**D**)
*Figure 7. Continued on next page*

*Figure 7. Continued*

shows the expression of Notch (white) in myoblasts lineage (green) along with Numb (red). (**E**) Quantification of number of PH3 positive AMPs in Numb knock down (*Dmef2*-Gal4, TubGal80ts > UAS *Numb* RNAi) showing significant increase in the total proliferating AMPs. Gal80 repression was relieved from early third instar till late third instar by shifting from 18°C to 29°C. All graphs are Mean ± Standard Error (Student's *t* test). n = 10. p-values < 0.001. (**F–K**) Twinspot MARCM in Numb RNAi (**F–H**) and in Notch upregulation (**I–K**) backgrounds, generated in early third instar, showing loss of asymmetry or symmetric clones in contrast to wild type asymmetric clone as shown in *Figure 2D–F*. n = 5. Scale bar, 50 μm.

(Ubx) is removed from the haltere disc, showed that AMP population is controlled by the disc-epidermis. When the haltere ectoderm is transformed to T2 with the AMPs continue to retain a T3 identity, the AMPs nevertheless transform to a T2 population size (*Fernandes et al., 1994*; *Roy et al., 1997*). Despite these early studies examining the role of signaling from the disc-ectoderm (*Sudarsan et al., 2001*) in flight muscle specification (*Schnorrer et al., 2010*; *Schönbauer et al., 2011*), the mechanism of population expansion have not been hitherto investigated. This is an important general problem as it addresses how diversification of internal tissues in segmented animals takes place.

Our results show that, in early development the AMPs on the wing-disc go through a phase of exponential amplification from about 10 cells to 250. This amplification, where the daughters of each stem are both stem cells, takes place by the division of cells parallel to the disc-epithelium stem-cell niche and requires Ser-N signaling. As the expression of Wg in the presumptive notum known to occur in the late second instar larva (*Tomoyasu et al., 2000*; *Alexandre et al., 2014*) changes in the AMPs proliferative behaviour are seen. A switch to asymmetric cell- division results in a 'self-renewed' stem cell and a post-mitotic sibling poised to differentiate to contribute to multinucleate muscles. The plane of cell division is no longer parallel to the ectoderm, resulting a stratified layer of cells, with the disc-proximal layer consisting of stem cells.

Our findings support a model in which the embryonically generated AMPs acting as postembryonic muscle stem cells generate adult-specific myoblasts in a two-step process (*Figure 10*). AMPs initially manifest a symmetric cell division mode, which serves to amplify the progenitor pool in the first and second larval stages, and subsequently transit to an asymmetric cell division mode during the third larval stage during which they self-renew and generate the postmitotic myoblasts required for adult muscle formation. In both steps, signaling is required from the notum region of the wing disc epithelium, which acts as a transient niche. Thus, Ser localized in epithelial cells of the disc is necessary for activation of N signaling in AMPs throughout larval development, and diffusible Wg from the disc epithelial cells in the third larval instar is necessary for the expression of Nb in AMP lineages and, hence, for their transition from symmetric to asymmetric division modes. The idea of epithelium tissue as transient niche for the regulation of proliferation is something similar to that known in intestinal stem cell (*Mathur et al., 2010*). In this paper authors provide evidence for the role of progeny of ISC progenitor as a very transient signaling center regulating stem cell proliferation via decapentaplegic.

Since these findings reveal remarkable parallels to other tissue stem cell systems in *Drosophila*, we consider the disc-associated AMPs to be a novel type of muscle stem cell that orchestrates the early phases of adult flight muscle development in *Drosophila*. This is a newly identified and key role for Wg in the presumptive notum. Its other roles include a requirement for expression of the muscle-attachment gene *stripe* (*sr*), in the presumptive notum (*Ghazi and VijayRaghavan, 2003*) and for maintenance of the indirect-flight muscle marker Vg, in AMPs (*Sudarsan et al., 2001*).

Wg causes this switch in cell- division pattern by the transcription of Nb in the stem-cell layer, and its asymmetric distribution by well-established mechanisms (*Skeath and Doe, 1998*; *Couturier et al., 2012*) thereby inhibiting Notch activation in the post mitotic progeny.

Many types of tissue stem cells, in a manner similar to that shown here for the AMPs, have the capability of proliferating through symmetric cell divisions, thereby expanding the stem cell pool, and then switching to asymmetric cell divisions. In *Drosophila* optic lobe development, a set of 30–40 epithelial progenitor cells generated in the embryo begin to proliferate in a symmetric cell division mode during early larval development to generate a neuroepithelial cell pool of approximately 700 cells. These neuroepithelial cells sequentially transform into neuroblasts, which generate differentiated neural cells of the medulla through asymmetric divisions. This transition from symmetric to asymmetric divisions involves cellular events such as the reorientation of the mitotic spindle and multiple molecular signaling processes including the dynamic regulation of Notch signaling (*Egger et al., 2010, 2011*). Since this transition from symmetrical 'proliferative' divisions to asymmetrical 'differentiative' divisions

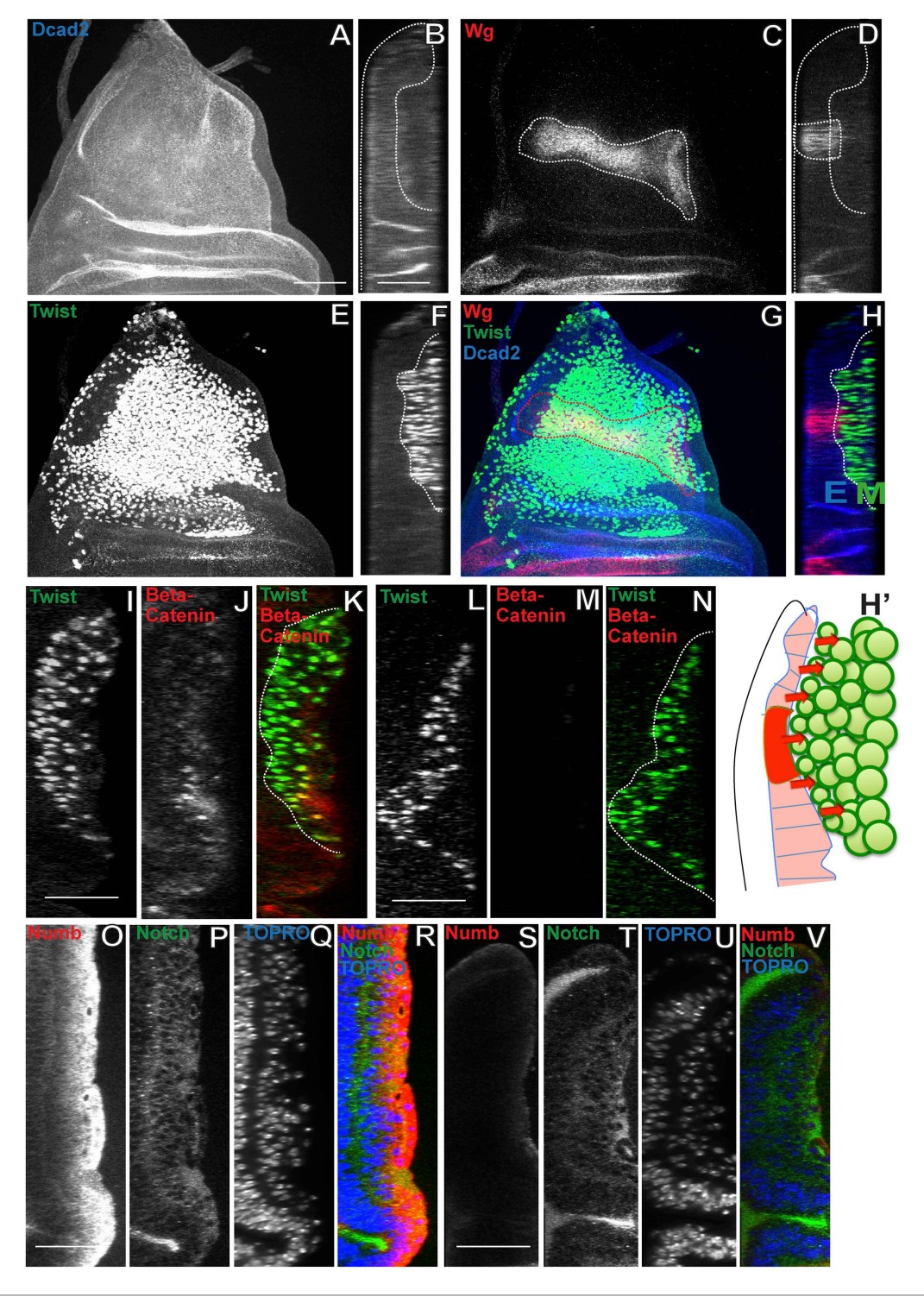

**Figure 8**. Wingless signaling from the wing disc epithelium induces Numb expression in third instar AMP lineages. (**A**–**H**) Third instar wing disc stained for Dcad2 (anti-Dcad2, Blue), Wg (anti-Wg, red) and Twi (anti-Twist, green) demonstrating a prominent longitudinal stripe of Wg expression in disc epithelium. n = 20. (**H′**) Schematic of the merge (**H**) depicting disc epithelium, as a source of Wg production and dispersal subsequently leading to Wg signaling (red arrows) activated in all AMPs. (**I**–**N**) AMP lineages (anti-Twi, green) stained for Beta-catenin (down stream molecule of Wg pathway) (anti-Beta catenin, red) in Canton-S (**I**–**K**) and in *Wg(ts)/Wg(Sp-1)* alleles (loss of

*Figure 8. Continued on next page*

*Figure 8. Continued*

function alleles of Wg gene) (**L–N**). Loss of Beta-catenin in *Wg(ts)/Wg(Sp-1)* shows absence of Wg activation in AMPs which also leads to decrease in number of AMP lineages. Presence of Beta-catenin in all AMPs clearly points towards Wg action at a distance far from the disc epithelium, the source of Wg. n = 10. (**O–R**) The optical section of third instar wing disc showing Numb expression (anti-Numb, red) very prominent in the distal layer of AMPs, marked by Notch (anti-Notch, green) (also in *Figure 5A–C*). (**S–V**) Wg loss of function (*Wg(ts)/Wg(Sp-1)*) results in total disappearance of Numb in AMPs. n = 10. Scale bar, 50 µm.

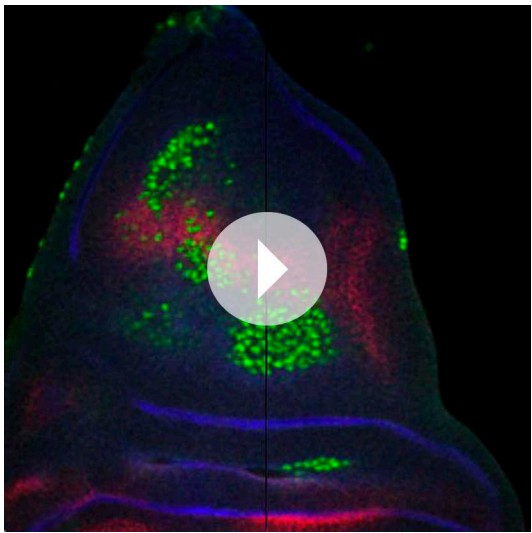

**Video 5**. Wg staining in Canton-S (related to *Figure 8*). Late third instar wing disc showing Wg (anti-Wg, red) expression in disc epithelium (anti-Dcad2, blue). The myoblasts forms a close association with Wg expression.

is also manifest in neural stem cell pools of the developing retina and neural tube in vertebrates, it may be a general strategy for generating a large final number of differentiated progeny from a small initial number of progenitors (*Kriegstein et al., 2006*; *Borrell and Reillo, 2012*). The *Drosophila* adult intestine and that of the mouse are other examples of use of both modes of cell amplification in response to adult environmental inputs such as diet and consequent insulin-pathway signaling (*Lopez-Garcia et al., 2010*; *O'Brien et al., 2011*; *Yilmaz et al., 2012*).

Our results are consistent with the population of cells proximal to the epithelium being stem cells: These disc-proximal cells self-renew and also give rise to siblings, which differentiate. However, as of now, we do not have a marker that specifically labels these proposed stem cells. In this situation our results are also consistent with the view that the myoblasts are a pool of cells, in which only those closest to the signaling center receive signals that impact their division.

One of the characteristics of the relationship between stem cells and their niche is the orientation of cell-division. This ensures the balance between self-renewal and differentiation as progeny, which lose contact with the signaling niche, is canalized to a particular fate, whereas those still in contact remain as self-renewing stem cells. In germline stem cell (GSC) in the ovarioles and testis, the contact of stem cells with the tightly regulate stem cell number and coordinated maturation of the progeny to form gametes. Loss of regulation in this axis results in mis-regulated cell number amplification (*López-Onieva et al., 2008*; *Wang et al., 2008*; *Losick et al., 2011*). The intestinal stem cell niche in *Drosophila* and *Mus* have a cryptic stem cell niche where a differentiating progeny signal back to the stem cell through EGF signaling, acting as mitogen (*Jiang and Edgar, 2009*). Neuroblasts in *Drosophila* are widely considered as neural stem cells as they are shown to go through a series of self-renewing divisions while siblings differentiating to form neurons. Remarkably, this is also seen in the culture, suggesting that this is an intrinsic property and not niche dependent. However, surrounding glia in vivo are proposed to have a regulatory effect on the stem cell abilities of these cells (*Chai et al., 2012*).

There have also been previous studies in *Drosophila* that address how stem cell division relates to regulation of population size. It is known from the studies in germline stem cell (*Gilboa and Lehmann, 2006*) and neural stem cells (*Egger et al., 2011*) there is change in division rates during larval development. In the early instars there is an amplification that slows down in the third instar. The ovary of adult *Drosophila* has 16–18 units, ovarioles, which are formed during the larval stages. Each ovariole contains two or three germline stem cells, which are in contact with somatic cells that regulate their establishment, maintenance and differentiation. For example, the developmental origin of the ovaries are the primordial germ cells (PGCs), which till the third instar contribute to most of the cells required for gonad formation and in pupal stage separate into ovarioles (*Gilboa and Lehmann, 2006*). The numbers of PGCs double every 24 hr after hatching till the mid-third instar after which it slows down, finally forming 100 cells starting from 12 cells (*Gilboa and Lehmann, 2006*). Such examples and our

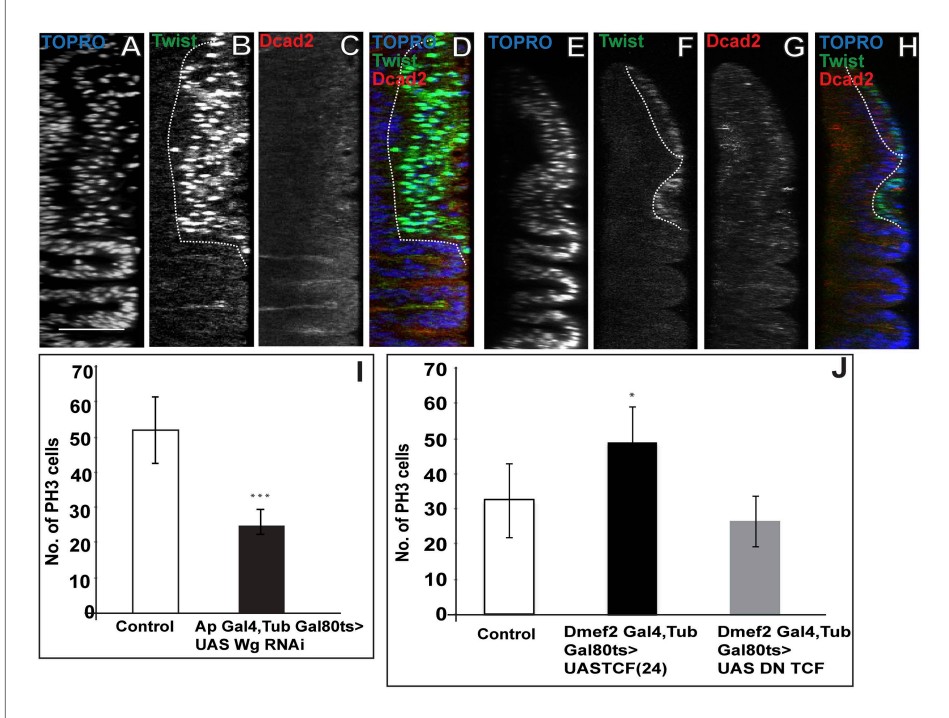

**Figure 9**. Loss of Wg results in reduction of mitotic activity and layered arrangement in AMP lineage. (**A–H**) Third instar wing disc showing loss of multilayered arrangement in Wg loss of function back ground (*Wg(ts)/Wg(Sp-1)*). In control disc (Canton-S) the multistratified arrangement can be distinctly seen which clearly disappears in Wg loss of function background. n = 8. (**I**) Quantification of number of PH3 positive AMPs in epithelium specific Wg knockdown using *Ap*Gal4 > UAS *Wg* RNAi showing significant decreases in comparison to control (Canton-s). All graphs are Mean ± Standard Error (Student's *t* test). p-value < 0.001, n = 10. (**J**) AMP specific perturbations (using *Dmef2*-Gal4) of Wg pathway downstream molecules (TCF) showing changes in mitotic activity in comparison to control. The activation of Wg pathway by overexpressing activated TCF (UAS *TCF* 24) leads to significant increase in mitotic activity. Gal80 repression was relieved from early second instar till late third instar by shifting from 18°C to 29°C. All graphs are Mean ± Standard Error (Student's *t* test). n = 15.

The following figure supplements are available for figure 9:

**Figure supplement 1**. Membrane tethered Wg perturbs multistratified arrangement of AMPs.

**Figure supplement 2**. Wg down regulation alters division axis of third instar AMPs.

results tempt us to speculate that an early amplification and a later slowing down maybe a general feature of regulation of stem cell proliferation in *Drosophila*.

Our results and the underlying mechanisms could also be of general applicability in understanding myogenesis in other contexts. In vertebrates, the Hox identity of the ectoderm could influence proliferation in the mesoderm, in a manner similar to that observed by us, to create the appropriate population of progenitors for the morphogenesis of muscles of very different sizes. Our study suggests the testable hypothesis that vertebrate myoblast pools also develop by symmetric and asymmetric division of muscle progenitor stem cells.

In vertebrate muscle, satellite cells form a quiescent muscle stem-cell population important for the regeneration (***Brack and Rando, 2012***; ***Cooper et al., 1999***; ***Le Grand and Rudnicki, 2007***; ***Vasyutina et al., 2007***). A recent study (***Konstantinides and Averof et al., 2014***) of limb regeneration in the crustacean, *Parhyale hawaiensis* identified satellite like cells (SLCs) expressing Pax3/7 genes expressed in vertebrate muscle satellite cells (***Kassar-Duchossoy et al., 2005***). This study along with others show a common mesodermal origin of adult muscle stem cells in crustaceans and chordates though the precise developmental origins of these remains a mystery (***MAURO, 1961***; ***Bryson-Richardson and Currie, 2008***; ***Sambasivan and Tajbakhsh, 2007***). Our study raises the interesting possibility that

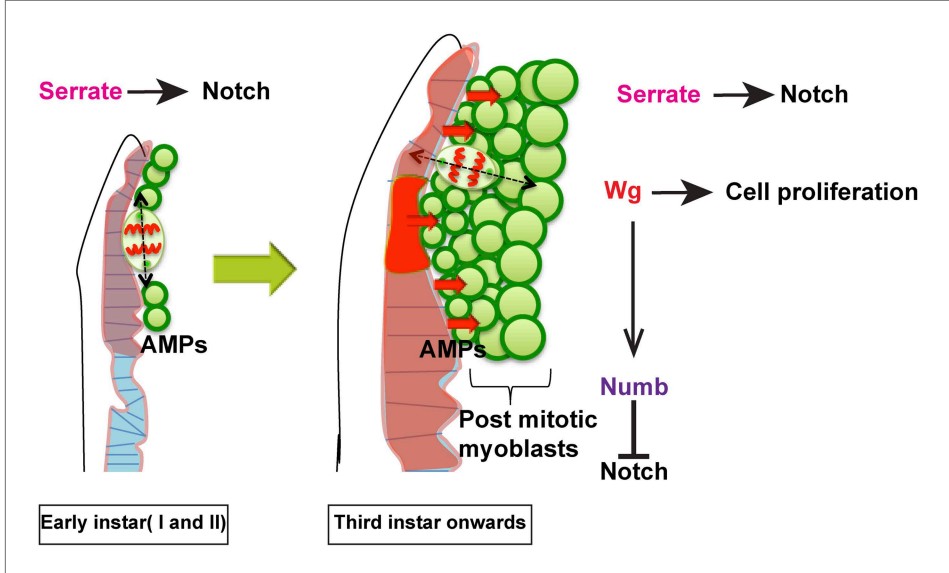

**Figure 10**. Model proposed. In the early instars (I and II) AMPs exhibit symmetric division along epithelium and Serrate-Notch signaling plays major role at this stage. In the third instar onwards the axis of cell division in AMPs changes to orthogonal orientation and expression of Wg in disc epithelium along with Serrate-Notch signaling regulates AMP proliferation. Wg signaling potentially regulates Numb, which inhibits Notch leading to asymmetric divisions.

muscle satellite cells are 'unfused' stem cells kept aside during early development. We also speculate that the novel stem cells that we observe could be satellite cells of flight muscle, used for repair upon muscle damage in a manner similar to that seen in vertebrates.

## Materials and methods

### Fly strains and genetics
Fly stocks were obtained from the Bloomington *Drosophila* Stock Centre (Indiana, USA) and, unless otherwise stated, were grown on standard cornmeal medium at 25°C. The following transgenic lines were used.

For wild type Twin-spot MARCM experiments males of genotype *+; FRT 40 A, UAS mCD8::GFP, UAS rCD2 RNAi; Dmef2*-Gal4 were crossed to females *Hsflp/Hsflp; FRT 40A, UAS rCD2::RFP, UAS GFP RNAi*.

For Notch over expression and Numb downregulation twinspot MARCM experiments males of genotype *+; FRT 40, UAS mCD8::GFP, UAS rCD2 RNAi; Dmef2*-Gal4 were crossed to females of genotype *Hsflp/Hsflp; FRT 40A, UAS rCD2::RFP, UAS GFP RNAi/Cyo Act-GFP; UAS NICD/TM6* or *Hsflp/Hsflp; FRT 40A, UAS rCD2::RFP, UAS GFP RNAi/Cyo Act-GFP; UAS Numb RNAi/TM6 Tb* respectively.

For MARCM experiments females of genotype *Hsflp/Hsflp; FRT 42 B, Tub Gal80* were crossed to males of genotype *+; FRT 42 B UAS mCD8::GFP/Cyo Act-GFP; Dmef2*-Gal4.

In knockdown and overexpression experiments:

*+; +; Dmef2*-Gal4, *Gal80ts. UAS Notch RNAi* (Bloom, 35213). *UAS Numb RNAi* (Bloom, 35045). *UAS Wg RNAi* (13352; VDRC, Austria). *UAS Serrate RNAi* (108348; VDRC, Austria). *UAS NICD. UAS DN Notch. UAS DN TCFΔN. UAS (TCF) 24/Cyo GFP.*

In mitotic spindle orientation experiments females of genotype *+; +; Dmef2*-Gal4 were crossed to males of genotype *P{UASp-GFP-Cnn1}26-1, w1118*.

### Other stocks
*Serrate lacZ9.1* (*Bachmann and Knust, 1998*). *Wg Sp-1/Cyo GFP. Wg^{ts}. Notch^{ts}.* wg{KO, Nrt-Wg} (*Alexandre et al., 2014*).

### Immunohistochemistry and microscopy
Wing discs were dissected from first instar (24–30 hr AEL), second instar (48–55 hr AEL) and third instar larval (72 hr onwards) stages and then fixed in 4% paraformaldehyde diluted in Phosphate

buffered saline (PBS pH-7.5). Immunostaining was performed according to *Ghazi et al. (2000)*. In brief, samples were then subjected to two washes of 0.3% PTX (PBS + 0.3% Triton-X) and 0.3% PBTX (PBS + 0.3% Triton-X + 0.1 %BSA) for 15 m each. Primary antibody staining was performed for overnight at 4°C on shaker and secondary antibodies were added following four washes of 0.3% PTX. Excess of unbound secondary antibodies were removed by two washes of 0.3% PTX following which samples were mounted in Vectashield mounting media. For immunostaining Anti Wg (Mouse, 1:100, DSHB), Anti-Twist (Rabbit, 1:100, kindly provided by S Roth, University of Cologne), Anti-NICD (Notch intracellular C-terminal domain) (Mouse, 1:100, DSHB), Anti-Numb (Rabbit, 1:100, kindly provided by Juergen Knoblich, IMBA, Vienna), Anti-GFP (Chick, 1:500, Abcam, Cambridge, UK), Anti-CD2 (Mouse, 1:100; Serotec, Raleigh, NC, USA), TO-PRO-3-Iodide (1:1000, Invitrogen), Anti-DCAD2 (Rat, 1:200, DSHB), Anti-Beta Gal (Mouse, 1:50, DSHB), Anti-Phospho histone 3 (Rabbit, 1:100, Millipore). Secondary antibodies (1:500) from Invitrogen conjugated with Alexa fluor-488, 568 and 647 were used in all staining procedures. Olympus FV 1000 confocal point scanning microscope was used for image acquisition, which were processed using ImageJ software.

Quantification of division axis was essentially performed as explained in the *Egger et al. (2007)*.

### Twin-spot MARCM and MARCM experiments

To generate twin-spot MARCM clones, a single heat shock of 15 m at 37°C was given to specifically staged larvae and then larvae were dissected and wing discs were removed. The samples were then processed for antibody staining as mentioned earlier. ImageJ software was used to determine the number of cells in each clone (Rasband WS, ImageJ U S. National Institutes of Health, Bethesda, Maryland, USA, http://imagej.nih.gov/ij/, 1997–2012).

### Edu labeling experiments

For Edu labeling, larvae were aged for 24, 48 and 72 hr after hatching, on standard cornmeal media and then pulse labeled for 5 hr on Edu (0.2 mM final concentration) mixed cornmeal media (*Daul et al., 2010*). Half of the larvae from the 72 hr stage were separated, dissected and processed for immunolabeling in the 'no chase' cohort. Remaining larvae were transferred to standard cornmeal media without Edu and allowed to develop until wandering third instar stage. Wing discs were then dissected and processed for immunolabeling studies. In both cohorts, Edu detection was performed according to the Click-iT Edu labeling kit (Invitrogen).

## Acknowledgements

This work was possible due to the generous support from the Tata Institute of Fundamental Research and the National Centre for Biological Sciences. We thank the Centre for Nanotechnology, NCBS (Department of Science and Technology Grant No. SR/S5/NM- 36/2005), for the Olympus FV1000 microscopes in the Central Imaging and Flow facilities (NCBS). Major additional support came from grants from the Department of Biotechnology, the Department of Science and Technology, Government of India and the Swiss SNF. We thank the *Drosophila* community for generous supply of fly strains and antibodies. Special thanks to Juergen Knoblich and colleagues (IMBA, Austria) for the anti-Numb antibody.

## Additional information

#### Competing interests

KVR: Senior editor, *eLife*. The other authors declare that no competing interests exist.

#### Funding

| Funder | Author |
| --- | --- |
| Department of Science and Technology, Ministry of Science and Technology | Rajesh D Gunage, K VijayRaghavan |
| Swiss National Science Foundation | Heinrich Reichert |

The funders had no role in study design, data collection and interpretation, or the decision to submit the work for publication.

## Author contributions

RDG, Conception and design, Acquisition of data, Analysis and interpretation of data, Drafting or revising the article; HR, KVR, Conception and design, Analysis and interpretation of data, Drafting or revising the article

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
