## [Decision Letter]

Thank you for sending your work entitled “Identification of a new stem cell population which generates *Drosophila* flight muscles” for consideration at *eLife*. Your article has been favorably evaluated by Janet Rossant (Senior editor) and 3 reviewers, one of whom, Duojia Pan, is a member of our Board of Reviewing Editors.

The Reviewing editor and the other reviewers discussed their comments before we reached this decision, and the Reviewing editor has assembled the following comments to help you prepare a revised submission.

In this study, Gunagee et al. investigated the growth and specification of adult muscle progenitor (AMP) cells of the large *Drosophila* indirect flight muscle. These cells are set aside during embryogenesis and undergo extensive proliferation in larval development to generate ∼2500 myoblasts associated with the wing-disc in the mesothoracic segment. They authors show that the AMPs undergo a series of symmetric divisions, regulated by Serrate-Notch signaling, resulting in their expansion. This is followed by a series of asymmetric divisions, regulated by Wg and numb, leading to presumably muscle cells. Their data therefore implicate the epidermal tissue of the wing imaginal disc as a stem cell niche that orchestrates the proliferation and differentiation of the AMPs.

Overall, this study is well conducted and should be of interest to the readership of *eLife*. The reviewers raised several questions that the authors may consider for improvement.

1) A direct implication of the authors' model, which was not tested in the paper, is that loss of wingless from the wing disc should change the asymmetric division pattern of AMPs, which can be revealed by the two-color MARCM analysis as the authors have done for examining Serrate-Notch-signaling. This experiment would greatly strengthen the authors' model concerning the role of Wingless in the switch to asymmetric division.

2) In Figure 3, the authors showed that symmetric and asymmetric division phases correlate with orientation of the mitotic spindle. What the frequencies of the type of division shown here? Are 3E-G (similarly, 3I-K) derived from the same sample or different samples? Also, rather than showing one example, it will be more informative to show the distribution of division planes relative to the wing disc epithelium from multiple examples, akin to the “pie chart” that is often used in studies of neuroblast asymmetric divisions (for example, see papers from Chris Doe's lab).

3) Error bars are missing in some figures or panels, such as Figure 1, Figure 2. Does this represent one wing disc? If not, how many samples are quantified in these experiments?

4) The authors write “Symmetric divisions result in daughter cell clones of RFP and GFP of equal number of cells; asymmetric cell divisions result in daughter cell clones of GFP and RFP of unequal size.” This claim is flawed. A clone could contain equal numbers of GFP and RFP cells and result from an asymmetric division. You really need a marker that is differentially expressed to get an idea about the nature of the division. Having said that, finding a marker differentially expressed could be misleading as the division could be symmetric and the placement of the daughter cells in different microenvironments could result in different expression patterns leading one to falsely conclude the division was asymmetric. Furthermore, a division could be symmetric and lead to unequal numbers of GFP and RFP cells. For example if one of the daughters moved into a microenvironment that was less proliferative than its sib. The authors should re-state the conclusion here: maybe it suffices to say that the division outcome is asymmetric.

5) One concern is the data assessing the role of Wg in controlling numb expression and Wg signaling in general. Firstly, in the absence of Wg signaling, Numb accumulation is lost from both ectodermal and myoblast cells (Figure 8), suggesting that Wg affects notum cells as well. A more major concern, however, is how Wg signaling can impact, in a similar manner, cells close to and distant from the signal. Beta-catenin and Numb expression are lost from myoblasts located a significant distance from the source of the Wg signal in the wg mutants. Is there a precedent for Wg signaling to extend over such large distances? In addition, there is no evidence in the data presented that beta-catenin levels, or Numb expression, differ whether the cells are closer to, or further away from, the source of the signal. These issues should be resolved before the mechanistic role of Wg can be included in the current model for how myoblast proliferation and behavior is controlled; otherwise the Wg studies could be removed from the manuscript.

---

## [Author Response]

*1) A direct implication of the authors' model, which was not tested in the paper, is that loss of wingless from the wing disc should change the asymmetric division pattern of AMPs, which can be revealed by the two-color MARCM analysis as the authors have done for examining Serrate-Notch-signaling. This experiment would greatly strengthen the authors' model concerning the role of Wingless in the switch to asymmetric division*.

The reviewers are correct that we have not explicitly demonstrated that loss of Wingless from the wing disc causes change in the asymmetric division pattern of AMPs. While other data indirectly support this claim, we appreciate that showing this directly would greatly strengthen the proposed model. We have resolved this matter as described below and have modified the text accordingly and Figure 9—figure supplement 2. (The experiment suggested by the reviewer demands substantial genetic recombination experiments bringing Wg alleles with the TWINSPOT fly. Our experience with such recombination experiments using the TWINSPOT flies always resulted in the loss of the one of the fluorescent protein coding genes culminating in the loss of one color. While this was frustrating alternative approaches fortunately worked. )

We have performed an experiment to understand the division pattern when the Wg pathway is downregulated. In the wild type, in early instars, the division axis is parallel to the epithelium. In the third instar there is a change in the division axis, effectively perpendicular to the epithelium and the division outcome is asymmetric. We down-regulated Wg function by overexpressing DNTCF and show by using CNN-GFP that the axis of the division in this background is parallel to the epithelium, similar to that seen in the early instars. Wg, in the presumptive notum is expressed only from the late 2^nd^ instar (1), therefore this result shows that down-regulation of Wg prevents the switch in cell-division orientation from parallel to the epithelium to perpendicular to the epithelium. This result greatly strengthens the results already shown in Figures 8 and 9 that reduction of Wg signaling reduces Numb expression and also the layers of AMPs that come about as a consequence of asymmetric proliferation.

*2) In*
Figure 3*, the authors showed that symmetric and asymmetric division phases correlate with orientation of the mitotic spindle. What the frequencies of the type of division shown here? Are 3E-G (similarly, 3I-K) derived from the same sample or different samples? Also, rather than showing one example, it will be more informative to show the distribution of division planes relative to the wing disc epithelium from multiple examples, akin to the “pie chart” that is often used in studies of neuroblast asymmetric divisions (for example, see papers from Chris Doe's lab)*.

We are very grateful for this suggestion. We have reanalyzed the data for the angle of division and represented them in a pie-chart format that is now in Figure 3. This supports our earlier observations and the text has been changed accordingly.

*3) Error bars are missing in some figures or panels, such as*
Figure 1*,*
Figure 2*. Does this represent one wing disc? If not, how many samples are quantified in these experiments?*

The quantification is a representation of data from single MARCM event but the experiment to confirm this observation was performed many times (N=12).

*4) The authors write “Symmetric divisions result in daughter cell clones of RFP and GFP of equal number of cells; asymmetric cell divisions result in daughter cell clones of GFP and RFP of unequal size.” This claim is flawed. A clone could contain equal numbers of GFP and RFP cells and result from an asymmetric division. You really need a marker that is differentially expressed to get an idea about the nature of the division. Having said that, finding a marker differentially expressed could be misleading as the division could be symmetric and the placement of the daughter cells in different microenvironments could result in different expression patterns leading one to falsely conclude the division was asymmetric. Furthermore, a division could be symmetric and lead to unequal numbers of GFP and RFP cells. For example if one of the daughters moved into a microenvironment that was less proliferative than its sib. The authors should re-state the conclusion here: maybe it suffices to say that the division outcome is asymmetric*.

We agree with reviewers’ comment and appropriate changes have been made in the text to incorporate these alternative possibilities.

In the Results section, the title has been changed from “AMP proliferation involves initial symmetric and subsequent asymmetric division modes” to “AMP proliferation involves initial symmetric and subsequent asymmetric clonal amplification” and the following statement has been added:

“This division is different from the earlier mode, as the outcome is asymmetric and the majority of the clone is essentially due to proliferation of one of the cells.”

*5) One concern is the data assessing the role of Wg in controlling numb expression and Wg signaling in general. Firstly, in the absence of Wg signaling, Numb accumulation is lost from both ectodermal and myoblast cells (*Figure 8*), suggesting that Wg affects notum cells as well. A more major concern, however, is how Wg signaling can impact, in a similar manner, cells close to and distant from the signal. Beta-catenin and Numb expression are lost from myoblasts located a significant distance from the source of the Wg signal in the wg mutants. Is there a precedent for Wg signaling to extend over such large distances? In addition, there is no evidence in the data presented that beta-catenin levels, or Numb expression, differ whether the cells are closer to, or further away from, the source of the signal. These issues should be resolved before the mechanistic role of Wg can be included in the current model for how myoblast proliferation and behavior is controlled; otherwise the Wg studies could be removed from the manuscript*.

We are pleased that the reviewer has raised these important points regarding the possible role of Wg as a long-range signal. We have performed analysis of the numb expression and also done three additional experiments to test the range of Wg signaling.

The expression of Numb in the myoblast pool was quantified (using Image J). Expression is least in the layers close to epithelium (actively dividing cells) in comparison to layers farthest from the epithelium (Figure 9—figure supplement 1).

The following experiments were also done:

In order to understand the possible role of Wg as a secreted long-range molecule, we made use of the tethered Wg fly (1), where the secreted form of Wg is replaced by the tethered form. We find:

A) Loss of beta-catenin the AMPs in third instar wing discs in homozygous animals of the genotype *wg*{*KO; NRT–Wg*}. This loss in cells close to the epithelium but distal to the Wg source supports the function of Wg as a long-range signal (It is important to keep in mind that Wg expression in the presumptive notum begins only late second instar (1),

B) Significant reduction of mitotic activity in AMPs in comparison to controls in homozygous *wg*{*KO; NRT–Wg*} third instar discs. This is consistent with Wg what is seen when Wg knockdown by other methods.

C) Reduction in the total myoblasts pool in *wg*{*KO; NRT–Wg*} and in layered arrangement of myoblasts.

The images for this are provided in Figure 9—figure supplement 1 (Membrane tethered Wg perturbs multistratified arrangement of AMPs). The text has been modified accordingly.